# A Software Verification Method for the Internet of Things and Cyber-Physical Systems

**Yuriy Manzhos \* and Yevheniia Sokolova \***

Department Software Engineering and Busines, National Aerospace University "Kharkiv Aviation Institute", 61070 Kharkiv, Ukraine
\* Correspondence: y.manzhos@khai.edu (Y.M.); y.sokolova@khai.edu (Y.S.)

**Abstract:** With the proliferation of the Internet of Things devices and cyber-physical systems, there is a growing demand for highly functional and high-quality software. To address this demand, it is crucial to employ effective software verification methods. The proposed method is based on the use of physical quantities defined by the International System of Units, which have specific physical dimensions. Additionally, a transformation of the physical value orientation introduced by Siano is utilized. To evaluate the effectiveness of this method, specialized software defect models have been developed. These models are based on the statistical characteristics of the open-source C/C++ code used in drone applications. The advantages of the proposed method include early detection of software defects during compile-time, reduced testing duration, cost savings by identifying a significant portion of latent defects, improved software quality by enhancing reliability, robustness, and performance, as well as complementing existing verification techniques by focusing on latent defects based on software characteristics. By implementing this method, significant reductions in testing time and improvements in both reliability and software quality can be achieved. The method aims to detect 90% of incorrect uses of software variables and over 50% of incorrect uses of operations at both compile-time and run-time.

**Keywords:** cyber-physical systems; internet of things; software defect model; software quality; physical dimension; physical orientation; formal verification





## 1. Introduction

The Internet of Things (IoT) is a contemporary paradigm that comprises a wide range of heterogeneous inter-connected devices capable of transmitting and receiving messages in various formats through different protocols to achieve diverse goals, as noted by Bai Lan et al. [1]. Presently, the IoT ecosystem encompasses over 20 billion devices, each with a unique identifier that can seamlessly interact via existing Internet infrastructure, as noted in [2]. These devices have diverse areas of application, ranging from inside the human body to deep within the oceans and underground. The IoT refers to a network of physical devices, vehicles, buildings, and other items that are embedded with sensors, software, and other technologies to enable them to collect and exchange data. The main focus of the IoT is on enabling communication between these devices to enable automation and control.

CPS (cyber-physical systems) are similar to the IoT; however, CPS specifically refer to a system of physical, computational, and communication components that are tightly integrated to monitor and control physical processes. CPS typically involve a closed-loop feedback control system that involves sensors, actuators, and computational elements to continuously monitor and adjust physical processes in real-time.

CPS integrate physical components with software components, as noted by Buffoni et al. [3]. According to references [4,5], CPS are able to operate on different spatial and temporal scales.

Control systems coupled to physical systems are a common example of CPS, with applications in various domains such as smart grids, autonomous automobile systems,

medical monitoring, industrial control systems, robotics systems, and automatic pilot avionics. CPS are becoming data-rich, enabling new and higher degrees of automation and autonomy.

New, smart CPS drive innovation and competition in a range of application domains, including agriculture, aeronautics, building design, civil infrastructure, energy, environmental quality, healthcare, personalized medicine, manufacturing, and transportation.

Despite some similarities, the primary distinction between the IoT and CPS is their focus. CPS are mainly focused on controlling physical processes, while the IoT is primarily focused on communication and data exchange between physical devices. CPS are typically used in industrial and manufacturing settings, where they facilitate real-time control of physical processes. On the other hand, the IoT has a broader range of applications, including home automation, healthcare, transportation, and other domains, where it enables seamless communication and integration of smart devices.

With the ever-increasing number of IoT and CPS devices, the need for more functional and high-quality software has become even more pressing. According to industry estimates, the global IoT market reached $100 billion in 2017, and this figure is projected to soar to $1.6 trillion by 2025, as noted in [6]. In 2022, enterprise spending on the IoT experienced a significant increase of 21.5%, reaching a total of $201 billion. Back in 2019, IoT analytics had initially projected a spending growth of 24% for the year 2023. However, their growth outlook for 2023 has been revised to 18.5% according to [7], as shown in Figure 1.

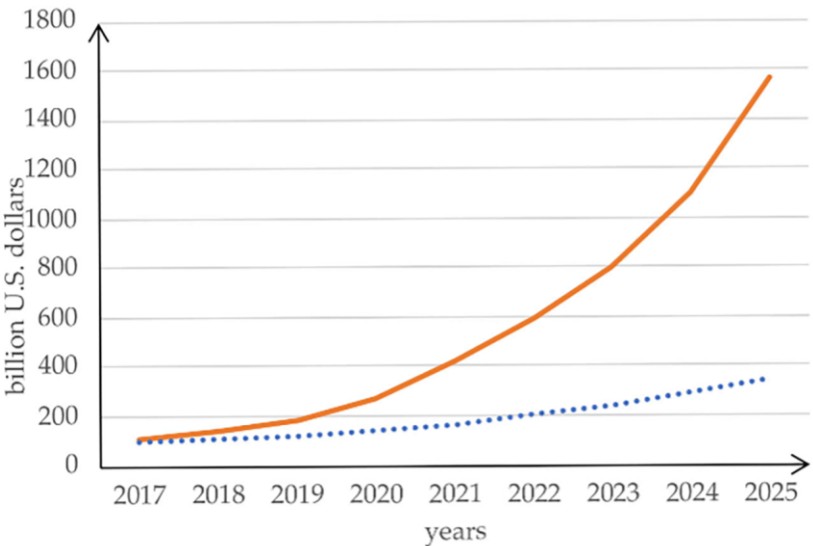

**Figure 1.** The global market of the Internet of Things (red solid line) and global spending on enterprise IoT technologies (blue dot line).

A model-based approach to CPS development is based on describing both the physical and software parts through models, allowing the whole system to be simulated before it is deployed.

There are several programming languages used in IoT and CPS development, including C/C++, Python, Java, JavaScript, and others. Among these, C/C++ is considered to be the most popular language for IoT development, with a popularity rate of 56.9%, according to recent research [8]. As of June 2023, GitHub has reported a total of over 53,285 IoT public repositories, with approximately 28,536 of them being C/C++ repositories, accounting for approximately 53.6% of the total number of IoT public repositories. Additionally, GitHub has reported over 22,776 CPS public repositories, out of which around 8892 are C/C++ repositories, making up approximately 39% of the total number of CPS public repositories. This is due to the fact that IoT devices typically have limited computing resources, and C/C++ is capable of working directly with the RAM while requiring minimal processing power.

CPS languages provide a unified approach to describing both the physical components and control software, making it possible to integrate modeling and simulation. Open standards such as FMI (Functional Mock-up Interface) and SSP (System Structure and Parameterization) facilitate this integration by defining a model format that utilizes the C language for behavior and XML for the interface. These standards, as specified in [9,10], enable the representation of pre-compiled models that can be exchanged between tools and combined for co-simulation.

According to the specifications outlined in [11], Modelica has been employed for the automatic generation of deployable embedded control software in C code from models. This utilization enhances the utility of Modelica as a comprehensive solution for the modeling, simulation, and deployment of CPS components.

The selection of a programming language for IoT and CPS development is highly dependent on the specific requirements of the project as well as the developer's proficiency. According to [12], utilizing C++ in embedded systems can be an effective solution, even considering the limited computing resources of microcontrollers used in small embedded applications compared to standard PCs. The clock frequency of microcontrollers may be much lower, and the amount of available RAM memory may be significantly less than that of a PC. Additionally, the smallest devices may not even have an operating system. To achieve the best performance, it is essential to choose a programming language that can be cross-compiled on a PC and then transferred as machine code to the device, avoiding any language that requires compilation or interpretation on the device itself, as this can lead to significant resource wastage.

For these reasons, C or C++ is often the preferred language for embedded systems, with critical device drivers requiring assembly language. If you follow the proper guidelines, using C++ can consume only slightly more resources than C, so it can be chosen based on the desired program structure. Overall, choosing the appropriate language for embedded systems can make a significant impact on performance and resource utilization.

The increasing number of IoT and CPS devices has resulted in a growing need for software that is both highly functional and of the utmost quality. As these devices become more ubiquitous and seamlessly integrated into our daily lives, the demand for dependable and efficient software becomes more critical than ever before. As a result, developers are constantly striving to enhance their software development methodologies and technologies to meet the ever-evolving demands of the IoT and CPS landscape.

However, given the increasing importance of the IoT and CPS as emerging technologies, it is expected that there will be more literature available on the topic of IoT and CPS software verification and quality assurance.

The typical software development life cycle (SDLC) involves several steps, including requirement analysis, design, implementation, testing and verification, and deployment and maintenance. While testing can increase our confidence in the program's correctness, it cannot prove it definitively. To establish correctness, we require a precise mathematical specification of the program's intended behavior and mathematical proof that the implementation meets the specification.

IoT verification encompasses a range of testing methodologies. These include conformance testing, as highlighted by Xie et al. [13], randomness testing, as discussed by Parisot et al. [14], statistical verification, as explored by Bae et al. [15], formal verification, as studied by Silva et al. [16], and the method known as model-based testing, as outlined by Ahmad [17]. In the specific context of the IoT, the model-checking technique, as emphasized by Clarke et al. [18], has notable representatives closely associated with it.

However, such software verification is difficult and time-consuming and is not usually considered cost-effective. In addition, modern verification methods would not replace testing in SDLC because most programs are not correct initially and need debugging before verification. The primary principle of verification involves adding specifications and invariants to the program and checking the verification conditions by proving generated lemmas based on the requirement specifications, as noted by Back [19].

However, most existing verification tools cannot detect software errors arising from incorrect usage of dimensions or units, which are commonly referred to as dimensionality errors or unit errors. These errors occur when software code or algorithms manipulate data with incompatible dimensions or units, resulting in incorrect calculations, unexpected behavior, or system failures. Such errors can have significant consequences across various domains, including engineering, finance, and scientific research:

1.  Inconsistent unit conversions
2.  Mixing incompatible units
3.  Incorrect scaling or normalization
4.  Mathematical operations on incompatible dimensions
5.  Inaccurate assumptions about input units

By being aware of these potential pitfalls and implementing proper checks, validation, and unit-aware programming techniques, developers can mitigate dimensionality or unit errors, ensuring accurate and reliable software functioning.

The failure of the Mars Climate Orbiter during its mission to study Mars' climate serves as a stark reminder of the consequences of navigational errors [20,21]. The spacecraft was intended to enter orbit around Mars in September 1999 but tragically entered the planet's atmosphere too low and disintegrated. This catastrophic error occurred due to a mismatch in the use of metric and imperial units, leading to incorrect calculations. Lockheed Martin, the contractor responsible for the spacecraft's navigation, used imperial units while NASA's software expected metric units. The failure resulted in a loss of $193.1 million and valuable scientific data. Lessons learned from this incident have since led to improved communication and unit conversion protocols in future space missions.

NASA's conversion concerns are particularly relevant to the constellation project, which places significant emphasis on manned spaceflight [22]. Launched in 2005, the project ambitiously aims to facilitate future moon landings. However, an obstacle arises as the project's specifications and blueprints are exclusively in British imperial units. The conversion of this extensive body of work into metric units poses a considerable estimated expenditure of approximately $370 million.

In 2003, Tokyo Disneyland's Space Mountain roller coaster experienced a disruptive event when it came to a halt due to a broken axle that failed to meet design requirements [23]. The axle's excessive gap, which exceeded 1 mm instead of the required 0.2 mm, led to fractures caused by vibrations and stress. Fortunately, no injuries occurred despite the derailment. The accident resulted from discrepancies in unit systems. In 1995, the coaster's axle specifications switched to metric units, but in August 2002, an order mistakenly reverted to British imperial units, leading to 44.14 mm axles instead of the required 45 mm ones.

In 1983, an Air Canada Boeing 767 experienced fuel depletion during a Montreal to Edmonton flight [24]. Low fuel pressure warnings at 41,000 feet led to engine failures. However, the skilled captain and first officer managed to land the plane safely at an unused air force base nearby, with only a few minor passenger injuries. The incident was caused by a malfunctioning fuel indication system and an incorrect density ratio of 1.77 pounds per liter instead of the correct 0.80 kg per liter. These factors led maintenance workers to manually calculate and pump less than half the required amount of fuel, contributing to the incident.

Adding to the list of errors, in the early 1990s during the creation of the "Mir" space orbital station, another incident occurred due to incorrect usage of units of measurement. When experts from the Moscow Design Bureau sent data in kilogram-force to Khartron in Kharkiv, Ukraine (where one of the authors of this article worked), it was mistakenly interpreted as newtons. Consequently, the control system of the module, weighing approximately 20 tons, had to be reprogrammed during the flight, leading to a two-week delay in its journey to the station.

To mitigate dimensionality or unit errors, it is crucial to follow best practices, which include the following:

1. Clearly specifying and documenting the expected units and dimensions of input and output data.
2. Implementing reliable and consistent unit conversion routines.
3. Leveraging libraries or frameworks with built-in support for units and dimensions.
4. Conducting comprehensive testing, including dedicated unit tests, to validate the accuracy of calculations and conversions.
5. Validating assumptions about input units and implementing suitable checks.
6. Providing informative error messages or warnings when dimensionality or unit errors are detected.

By being mindful of dimensions and units during software development, developers can reduce the occurrence of errors and ensure the accuracy and reliability of their software.

There are several libraries and frameworks available that offer built-in support for handling units and dimensions in software development. In the following are some popular options:

As noted by Matthias Christian Schabel et al. [25], Boost.Units provides a comprehensive framework for handling physical quantities in C++ programming. It allows you to work with quantities with different units, perform arithmetic operations, and ensure dimensional consistency at compile-time. Boost.Units offers compile-time dimensional analysis, type-safe unit conversions, and supports custom unit systems. It is particularly useful in scientific and engineering applications, where precise handling of units and dimensions is crucial for accurate calculations.

As noted by Edzer Pebesma [26], UDUNITS is a flexible and extensive library primarily used in scientific and meteorological applications. It offers a comprehensive database of physical units, allowing for unit conversions, arithmetic operations, and parsing of unit expressions. UDUNITS supports a wide range of units and provides bindings for various programming languages, including Python and Java. It is a reliable solution for managing units and dimensions, especially in domains that require extensive support for physical quantities.

According to [27], Units.NET is a powerful and user-friendly library for managing physical quantities in C# applications. It provides a comprehensive set of units, supporting unit conversions, arithmetic operations, and dimensional analysis. With Units.NET, developers can work with units and dimensions in a strongly typed manner, ensuring type safety and accurate calculations. It simplifies the handling of units and dimensions in C# projects, making it convenient to work with physical quantities.

Common disadvantages of the described libraries are that they cannot utilize orientational information for checking software code.

In addition, the utilization of a specialized software language called F# enables efficient manipulation of physical units and dimensions [28]. While F# is widely recognized for its applications in general-purpose programming and data analysis, it also proves to be highly effective in the context of the IoT (Internet of Things) and CPS (cyber-physical systems) domains.

However, in the case of reusing IoT and CPS software programs that employ different physical units and orientations of physical values, which are typically implemented in languages like C++, it becomes essential to undergo additional formal verification. This verification process should incorporate orientational and dimensional information to ensure successful integration and reduce the development time of new software projects.

This article focuses on exploring a formal verification method that utilizes dimensional and orientational homogeneities and natural software invariants. Specifically, it considers the dimensions and orientations of physical quantities as defined by the International System of Units (SI), as described in references [29,30]. Additionally, it incorporates transformations of physical quantity orientations proposed by Siano [31,32] and extended by Santos et al. [33]. By leveraging these invariants, this method can effectively verify the correctness of software and detect errors that may arise due to inconsistent or incorrect use

of units and dimensions, as well as incorrect usage of software operations, variables, and procedures, among other things.

The SI defines a set of base units, such as meters for length, kilograms for mass, and seconds for time, along with derived units, which are combinations of base units, such as meters per second for velocity or kilograms per cubic meter for density.

When it comes to software code, developers often need to work with and manipulate physical quantities in their programs. To ensure the correctness of such code, formal software verification methods can be applied. These methods use mathematical techniques to formally prove the properties of the software, such as its correctness, safety, or absence of certain errors.

The concept of homogeneity, derived from the SI system, plays a significant role in formal software verification. Siano proposed extending dimension homogeneity via orientation [31,32]. Now, the main concept of homogeneity states that any physical equation involving physical quantities must be both dimensionally and orientationally consistent. In other words, the units on both sides of an equation must match.

It is important to note that formal software verification involves more than just ensuring the homogeneity of physical quantities. It encompasses a broader range of techniques and methods to rigorously analyze and prove properties about software systems. However, leveraging the concept of homogeneity from the SI system can be a valuable tool in the pursuit of formal software verification, especially when dealing with physical quantities. The approach of leveraging the homogeneity of physical quantities and applying formal software verification techniques can make a significant contribution to ensuring functional and high-quality software for the IoT and CPS.

The advantages of the proposed method for software correctness and safety are that by utilizing formal software verification techniques, such as enforcing both dimensional and orientational homogeneity, developers can detect errors early in the development process and ensure that the software behaves as intended. This, in turn, reduces the potential for system failures or safety incidents.

IoT and CPS systems are typically subject to updates, maintenance, and evolution throughout their lifecycle. Enforcing both dimensional and orientational homogeneity and applying formal verification methods can enhance software maintainability and evolvability. By establishing clear and consistent units and enforcing them in the code, developers can more easily understand and modify the software, reducing the risk of introducing errors during updates or modifications. This promotes efficient maintenance and facilitates the evolution of the software as new requirements or functionalities are introduced.

Formal software verification methods, including the use of homogeneity, contribute to a rigorous quality assurance process. By systematically applying verification techniques, developers can identify and eliminate potential software defects, thereby improving the overall quality and reliability of IoT and CPS systems. This, in turn, enhances user satisfaction, reduces the risk of failures, and increases confidence in the deployed software. The thorough verification process helps ensure that the software meets the specified requirements and operates correctly in various scenarios, thereby ultimately enhancing the overall quality assurance efforts.

By combining the principles of homogeneity from the SI system with formal software verification methods, developers can create more robust, reliable, and functional software for the IoT and CPS. This approach helps mitigate risks, ensures safety, enhances interoperability, facilitates maintenance, and improves the overall quality of the software deployed in these systems.

## 2. The Formal Software Verification Method

Proposed is the utilization of natural software invariants, which are the physical dimensions and spatial orientation of software variables that correspond to real physical quantities. By incorporating these invariants into the program specification, it becomes

possible to convert all program expressions into a series of lemmas that must be proven. This enables the verification of the homogeneity and concision of the program.

### 2.1. Using Dimensional Homogeneity in Formal Software Verification

According to Martínez-Rojas et al. [34], dimensional analysis is a widely used methodology in physics and engineering. It is employed to discover or verify relationships among physical quantities by considering their physical dimensions. In the SI a physical quantity's dimension is the combination of the seven basic physical dimensions: length (meter, m), time (second, s), amount of substance (mole, mol), electric current (ampere, A), temperature (kelvin, K), luminous intensity (candela, cd), and mass (kilogram, kg). Derived units are products of powers of the base units, and when the numerical factor of this product is one, they are called coherent derived units. The base and coherent derived units of the SI form a coherent set designated as the set of coherent SI units. The word "coherent" in this context means that equations between the numerical values of quantities take the same form as the equations between the quantities themselves, ensuring consistency and accuracy in calculations involving physical quantities.

Some of the coherent derived units in the SI are given specific names and, together with the seven base units, form the core of the set of SI units. All other SI units are combinations of these units. For instance, plane angle is measured in radians (rad), which is equivalent to the ratio of two lengths; solid angle is measured in steradians (sr), which is equivalent to the ratio of two areas. The frequency is measured in hertz (Hz), which is equivalent to one cycle per second. The force is measured in newtons (N), which is equivalent to kg m/s$^2$. The pressure and stress are measured in pascals (Pa), which is equivalent to kg/m s$^2$ or N/m$^2$. The energy and work are measured in joules (J), which is equivalent to kg m$^2$/s$^2$ or N m.

The fundamental principle of dimensional analysis is based on the fact that a physical law must be independent of the units used to measure the physical variables. According to the principle of dimensional homogeneity, any meaningful equation must have the same dimensions on both sides. This is the fundamental approach to performing dimensional analysis in physics and engineering.

Existing software analysis tools only check the syntactic and semantic correctness of the code, but not its physical correctness. However, we can consider the program code of systems as a set of expressions consisting of operations and variables (constants). By using DA, we can verify the physical consistency of the program code and detect errors that may arise due to inconsistent or incorrect use of units and dimensions.

To check the correctness of expressions, we can use the dimensionality of program values. Preservation of the homogeneity of the expressions may indicate the physical usefulness of the expressions. Violation of homogeneity indicates the incorrect use of a program variable or program operation. Dimensional analysis provides an opportunity to check not only simple expressions but also calls to procedures and functions. The use of the physical dimension allows the verification of the software.

Dimensional analysis is a powerful tool that can be used to ensure the physical correctness of software code. By checking the dimensionality of program values, we can ensure the preservation of the homogeneity of expressions, which may indicate the physical usefulness of the code. In cases where homogeneity is violated, it may indicate an incorrect use of program variables or operations. Dimensional analysis can be applied not only to simple expressions but also to calls to procedures and functions, providing a comprehensive approach to verifying the physical consistency of the software.

We can view software as a model, and dimensional analysis can serve as a validation tool to ensure that this model adheres to the physical laws and principles governing the system it represents. By incorporating physical dimensions into the validation process, we can effectively identify and rectify errors that may arise from inconsistent or incorrect usage of units and dimensions. This approach ultimately contributes to the development of more reliable and accurate software.

In the software system, we can define it as a collection of interacting sub-systems. Each sub-system consists of interacting software units, and each unit comprises a set of operators. Operators, in turn, are ordered sets of statements or expressions. By structuring the software system in this hierarchical manner, we can effectively check the interactions and operations within the system.

To prove the homogeneity of software systems, we need to demonstrate the homogeneity of each subsystem. Similarly, to prove the homogeneity of subsystems, we need to establish the homogeneity of each software unit. Finally, to ensure the homogeneity of software units, we need to demonstrate the homogeneity of each software statement or expression. This stepwise approach allows us to systematically verify the homogeneity of the software and ensure its adherence to the specified physical dimensions.

Let us introduce a set of "multiplicative" operations {*, /, etc.} that generate new physical dimensions, while "additive" operations {+, −, =, <, ≤, >, ≥, !=, etc.} act as checkpoints to ensure dimensional homogeneity. If the source code contains variables that preserve specific physical dimensions, we can utilize this property, known as dimensional homogeneity, as a software invariant. Each additive operation serves as a source for generating lemmas. Following the principle of dimensional homogeneity, we can develop a set of lemmas to support the verification process.

By employing dimensional analysis, we can verify the physical dimensions of variables to identify errors resulting from inconsistent or incorrect usage of units and dimensions, as well as improper utilization of software operations, variables, and procedures. However, it is worth noting that certain variables may possess the same dimensions, such as moments of inertia and angular velocities. In order to detect software defects arising from the erroneous handling of such variables, careful examination of expressions involving angles, angular speed, and related quantities is required. It is important to remember that, according to the SI, angles are considered dimensionless values.

## 2.2. Using Orientational Homogeneity in Formal Software Verification

To address this issue, we can utilize features for transformations of angles and oriented values. In [31,32], Siano proposed an orientational analysis as an extension of dimensional analysis. This approach involves considering not only the physical dimensions but also the orientations of the quantities to enhance the analysis.

The use of orientational analysis can aid in expanding the base unit set while also ensuring dimensional and orientational consistency. Additionally, the orientational analysis technique can be applied for the formal verification of software code, allowing for a thorough evaluation of its accuracy and reliability.

Siano's proposed notation system for representing vector directions involved the use of orientational symbols $l_x$, $l_y$, $l_z$ [31,32]. Furthermore, a symbol without orientation represented by $l_0$ was introduced to represent vectors that do not possess a specific orientation.

For example, a velocity in the $x$-direction can be represented by $V_x \doteq l_x$, while a length in the $x$-direction can be represented by $L_x \doteq l_x$. Here, the symbol $\doteq$ denotes that the quantity on the left-hand side has the same orientation as the quantity on the right-hand side. In non-relativistic scenarios, mass is considered to be a quantity without orientation.

In order for equations involving physical variables to be valid, they must exhibit orientational homogeneity, meaning that the same orientation must be utilized on both sides of the equation. Furthermore, it is crucial that the orientations of physical variables are assigned in a consistent manner. For instance, the representation of acceleration in the $x$-direction as $a_x = \frac{\Delta V_x}{\Delta t}$, $V_x \doteq l_x$, $\Delta t \doteq l_0$ and $a_x \doteq \frac{l_x}{l_0}$ is only valid if both sides have the same orientation.

But what about the orientation of time? From the expression $H = \frac{gt^2}{2}$ we can define the time as follows: $t = \sqrt{\frac{2H}{g}} \doteq \sqrt{\frac{l_z}{l_z}} \doteq \sqrt{l_z l_z} \doteq \sqrt{l_0} \doteq l_0$.

The physical quantity of time is considered to be without orientation, meaning it does not possess a specific orientation in space.

In order to maintain orientational homogeneity, it is necessary to introduce a characteristic length scale $l_0$, since time is a quantity without orientation. Therefore, $l_x$ can be expressed as $l_x \doteq \frac{l_x}{l_o} \doteq l_x$ and $l_o l_x \doteq l_x l_o \doteq l_x$, $l_0^{-1} \doteq l_o$ ensuring that the orientation of $l_x$ remains consistent. That is why $a_x \doteq l_x$.

It is essential to assign orientations to physical variables in a consistent manner. For instance, pressure is defined as force per unit area. If the force is acting in the $z$-direction, then the area must be normal to it etc.:

$$P = \frac{F_z}{S_{xy}} \doteq \frac{l_z}{l_x l_y} \doteq \frac{l_z}{l_z} \doteq l_0, \ P = \frac{F_x}{S_{yz}} \doteq \frac{l_x}{l_y l_z} \doteq \frac{l_x}{l_x} \doteq l_0, \ P = \frac{F_y}{S_{xz}} \doteq \frac{l_y}{l_x l_z} \doteq \frac{l_y}{l_y} \doteq l_0.$$

In order for pressure to be considered a quantity without orientation, the area $S$ must have the same orientation as the force $F$. If the force $F$ is in a particular direction, then the area $S$ must be normal to that direction, meaning that both variables have the same orientation. Therefore, pressure can only be a quantity without orientation if this consistent orientation is maintained.

We define this as follows:

$$l_x l_y \doteq l_z, \ l_y l_z \doteq l_x, \ l_x l_z \doteq l_y, \ \frac{l_x}{l_x} \doteq l_0, \ \frac{l_y}{l_y} \doteq l_0, \ \frac{l_z}{l_z} \doteq l_0.$$

If $a \doteq l_x$, $b \doteq l_y$, $c \doteq l_z$ a volume of space, $V$, is a quantity without orientation:

$$V_{abc} = S_{ab}c \doteq l_z l_z \doteq l_0, \ V_{abc} = S_{bc}a \doteq l_x l_x \doteq l_0, \ V_{abc} = S_{ac}b \doteq l_y l_y \doteq l_0$$

Let us take a look at uniformly accelerated motion in the $x$-direction: $S_x = S_{0X} + v_x t + \frac{a_x t^2}{2}$, where $S_x$ is the total distance, $S_{0X}$ is the initial distance, $v_x$ is the velocity, and $a_x$ is the acceleration.

According to orientational homogeneity

$$S_x \doteq l_x, \ S_{0X} \doteq l_x, \ v_x t \doteq l_x l_0 \doteq l_x, \ \frac{a_x t^2}{2} \doteq \frac{l_x}{l_0}(l_0)^2 \doteq l_x$$

The orientation of derived physical variables, such as kinetic energy, can be determined by properly assigning orientation to primitive variables and applying the corresponding multiplication rules:

$$KE = \frac{mv_x^2}{2} + \frac{mv_y^2}{2} + \frac{mv_z^2}{2}, \ KE \doteq l_0 l_x l_x + l_0 l_y l_y + l_0 l_z l_z, \ KE \doteq l_0 l_0 + l_0 l_0 + l_0 l_0 \doteq l_0$$

We considered the orientation of an angle $\alpha$ in the $x$-$y$ plane. Because $\tan(\alpha) \doteq \frac{l_y}{l_x}$ and $\lim_{\alpha \to 0}(\tan(\alpha)) = \alpha$ we deduced that $\alpha \doteq \frac{l_y}{l_x} \doteq l_z$ and an angular velocity of $\omega_{xy} = \frac{\Delta \alpha}{\Delta t} \doteq l_z$.

Let us take the following series:

$$\text{sine}(\alpha) = \alpha - \frac{\alpha^3}{3!} + \frac{\alpha^5}{5!} \dots, \ \text{cosine}(\alpha) = 1 - \frac{\alpha^2}{2!} + \frac{\alpha^4}{4!} \dots$$

If $\alpha$ has any orientation, then sine $(\alpha)$ would also have that orientation, while the cosine $(\alpha)$ would be a quantity without orientation. This is because the sine function involves the odd powers of $\alpha$, while the cosine function involves the even powers of $\alpha$.

Siano demonstrated that orientational symbols have an algebra defined by the multiplication table for the orientation symbols [31,32], which is as follows:

$$
\begin{array}{c|cccc}
 & l_0 & l_x & l_y & l_z \\
\hline
l_0 & l_0 & l_x & l_y & l_z \\
l_x & l_x & l_0 & l_z & l_y \\
l_y & l_y & l_z & l_0 & l_x \\
l_z & l_z & l_y & l_x & l_0
\end{array}
\quad \text{and rules}: \quad
\begin{array}{ll}
l_o = \frac{1}{l_o} & l_x = \frac{1}{l_x} \\
l_y = \frac{1}{l_y} & l_z = \frac{1}{l_z}
\end{array}
$$

Based on the above, the product of two orientated physical quantities has an orientation as follows:

$$l_o l_x = l_x l_o = l_x, \; l_o l_y = l_y l_o = l_y, \; l_o l_z = l_z l_o = l_z, \; l_x l_x = l_y l_y = l_z l_z = l_0$$

If a source code contains variables that represent physical quantities with orientations, we can use the property of orientational homogeneity as a software invariant. By applying orientational homogeneity, we can transform the source code into a set of lemmas. Multiplicative operations, such as multiplication and division, introduce new physical orientations. On the other hand, additive operations such as addition, subtraction, and relational operators (e.g., =, <, $\leq$, >, $\geq$, !=) serve as checkpoints for verifying orientational homogeneity.

*2.3. Examples of Software Formal Verifications*

**Example 1.** *Consider the expression $F = ma$, where $F$ is a force with physical units of ($kg \, m \, s^{-2}$), $m$ is the mass in ($kg$), and $a$ is the acceleration in ($m \, s^{-2}$). This expression allows us to derive the orientation and dimension of the product $ma$. If $m$ is without orientation (denoted by $m \doteq l_0$) and $a$ is in the x-direction (denoted by $a \doteq l_x$), then $F$ has the x-orientation. If $m$ has units of ($kg$) and $a$ has units of ($m \, s^{-2}$), then the dimensions of the result are ($kg \, m \, s^{-2}$). The assignment operation "=", which is also known as the equality operator, acts as a checkpoint for our software invariants. It ensures that the physical orientation of $F$ is equal to the physical orientation of $ma$ and that the physical dimensions of $F$ are equal to the physical dimensions of the result.*

**Example 2.** *Consider the expression $S = S_0 + vt + 0.5at^2$, where $S$ represents the total distance, $S_0$ is the initial distance, $v$ is the velocity, $a$ is the acceleration, and $t$ is the time. This expression generates two new physical dimensions and orientations. The second "+" operation checks the dimensions and orientations of $vt$ and $0.5at^2$. The first "+" operation checks the homogeneity of $S_0$ and the result of the previous operation. Finally, the assignment operator "=" checks the homogeneity of $S$ and the result of the previous operation. By checking the homogeneity of these variables and operations, we can ensure that the physical dimensions and orientations are consistent throughout the expression.*

**Example 3.** *When calling procedures and functions, it is not always possible to check the physical dimensions and orientation of the arguments. However, for function signatures such as Type1 and some Function2 (Type2 x . . . ), where Type1 and Type2 have information about the physical dimensions and orientations of their arguments, it is possible to check the physical dimensions and orientations of the arguments. Each argument of a function generates a special lemma that can be used to prove dimensional and orientational homogeneities. Only after proving all the lemmas can we prove the correctness of the function call. It is important to note that real arguments of exponential and logarithmic functions must be dimensionless and without orientation to preserve dimensional and orientational homogeneities. On the other hand, arguments of trigonometric functions such as sine(x), cosine(x), and tan(x) must be orientated to preserve orientational homogeneity, while also being dimensionless to preserve dimensional homogeneity. The proposed method allows for the checking of physical dimensions and orientations in software statements and units. Repeating this check helps to ensure the correctness of the software system.*

**Example 4.** *Let us examine Euler's rotation equations, as described in [35], which find numerous applications in fields such as cyber-physical systems (CPS) and the Internet of Things (IoT). These equations are utilized in various contexts, including unmanned cars, helicopters, and other aerial vehicles.*

The general vector form of the equations is $I\dot{\omega} + \omega \times (I\omega) = M$, where $M$ represents the applied torques and $I$ is the inertia matrix. The vector $\dot{\omega}$ represents the angular acceleration.

In orthogonal principal axes of inertia coordinates the equations become

$$\begin{cases} I_x\dot{\omega}_x + (I_z - I_y)\omega_y\omega_z = M_x \\ I_y\dot{\omega}_y + (I_x - I_z)\omega_z\omega_x = M_y \\ I_z\dot{\omega}_z + (I_y - I_x)\omega_x\omega_y = M_z \end{cases} \tag{1}$$

where $M_x$, $M_y$, $M_z$ are the components of the applied torques (kg m$^2$ s$^{-2}$); $I_x$, $I_y$, and $I_z$ are the principal moments of inertia (kg m$^2$); $\omega_x$, $\omega_y$, and $\omega_z$ are the components of the angular velocities (s$^{-1}$); and $\dot{\omega}_x = \frac{d\omega_x}{dt}$, $\dot{\omega}_y = \frac{d\omega_y}{dt}$, and $\dot{\omega}_z = \frac{d\omega_z}{dt}$ have dimension (s$^{-2}$). However, $M_k$, $I_k$, $\omega_k$, and $\dot{\omega}_k$ have different orientations $l_k$, where $k = x, y, z$.

We can verify the dimensional homogeneity of Equation (1), but it is not possible to identify all defects. This is because certain quantities have the same dimensions and cannot be distinguished solely based on their units. For example, the dimensions of the moments of inertia ($I_x$, $I_y$, $I_z$) are (kg m$^2$) and angular velocities ($\omega_x$, $\omega_y$, $\omega_z$) are (s$^{-1}$), and the dimensions of the angular acceleration components ($\dot{\omega}_x$, $\dot{\omega}_y$, $\dot{\omega}_z$) are (s$^{-2}$) since angles are dimensionless. Therefore, while dimensional analysis can help identify some potential issues with the equation, it may not be able to catch all possible defects. For example, we cannot detect a defect if the expression $S = S_0 + vt + 0.5\,at^2$ does not include the initial position $S_0$.

In the context of Equation (1), the parameters ($M_x$, $M_y$, $M_z$, etc.) may have different orientations or values, which can help in detecting defects.

Furthermore, checking both the dimensional and orientational homogeneities of an equation can improve our ability to detect defects and ensure their correctness. This approach can be useful in the formal verification of CPS and IoT software, as it can help identify potential issues before they lead to real-world problems.

Let us assess the probability of detecting a software defect using both dimensional analysis and orientational analysis.

*2.4. Software Defect Detection Models*

2.4.1. General Software Defect Detection Model

To simplify the analysis, we assume that the software statement can only have one defect with a probability of $P_{def}$. The model starts with the initial event state of 'Software' and branches out into two possible outcomes at the next level: 'Software has a defect' and 'Software does not have a defect', with probabilities of $P_{def}$ and $1 - P_{def}$, respectively.

Decision trees, as described in [36], are visual representations utilized in decision analysis and machine learning. They illustrate decisions or events along with assigned probabilities or outcomes. Decision trees offer a structured approach to analyzing intricate decision-making processes. They can be applied to predict software defect detection, facilitating the identification and prevention of software issues, as demonstrated in Figure 2.

In the state 'Software has a defect', our focus shifts to detecting the defect. At the third level, the model branches out into two possible outcomes: 'Defect detected' and 'Defect not detected', with probabilities of $P_{DD}$ and $1 - P_{DD}$, respectively.

The revised sentence maintains clarity and correctness in grammar.

To define the conditional probability of software defect detection, we used the following formula:

$$\eta = \frac{P_{def} P_{DD}}{P_{def} P_{DD} + P_{def}(1 - P_{DD})} = P_{DD},$$

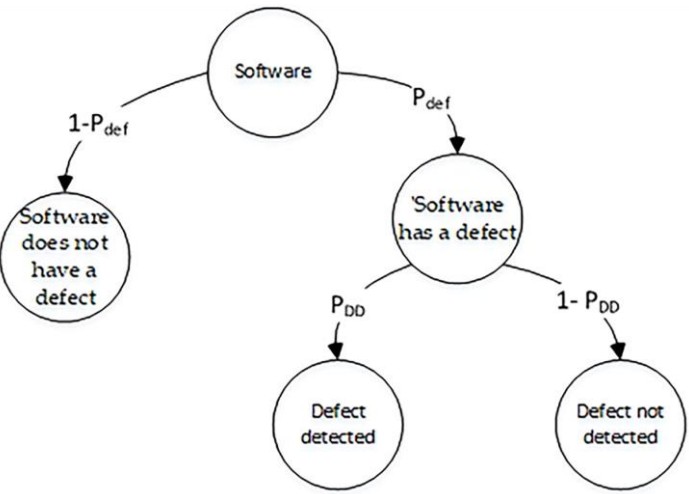

**Figure 2.** General software defect detection model.

Here $P_{def}$ represents a probability of a software defect in the code; $P_{DD}$ represents a probability of a software defect detection.

We can also introduce a more complex general software defect detection model (see Figure 3), which accounts for two types of defects: variable defects (uncorrected usage of a variable with incorrect dimension or orientation) and operation defects (incorrect usage of an operation). Despite the presence of multiple types of defects, the model still assumes that there is only one defect present in the software statement at any given time.

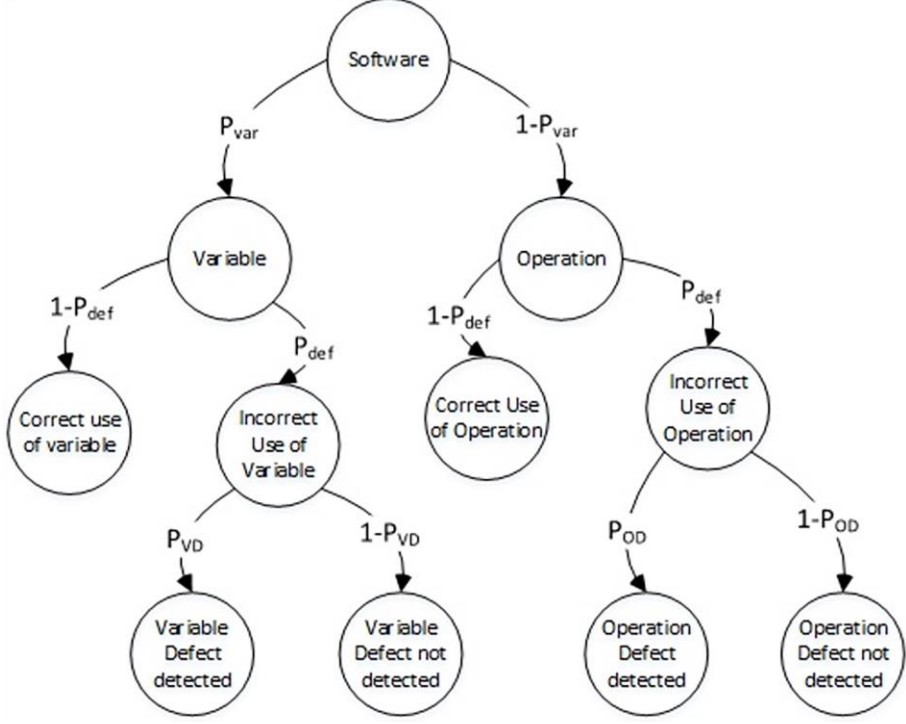

**Figure 3.** Complex general software defect detection model.

In this more complex model, the software statement can have two types of defects: variable defects and operation defects. A 'Variable defect' occurs when there is an incorrect usage of a variable in the code, such as using the wrong variable name. An 'Operation defect' occurs when there is an incorrect usage of an operation in the code, such as using the wrong operator symbol. Despite the presence of these two types of defects, the model still assumes that only one defect is present in the statement at any given time.

In this more complex model, the initial event state is 'Software'. At the branching point, the model expands into two possible outcomes: 'Variable' and 'Operation', with probabilities of $P_{\text{var}}$ and $1 - P_{\text{var}}$, respectively.

The 'Variable' state has two potential outcomes at the next level: 'Correct use of variable' and 'Incorrect use of variable', with probabilities of $1 - P_{def}$ and $P_{def}$, respectively.

The 'Incorrect use of variable' state then branches out into two possible outcomes at the next level: 'Variable defect detected' and 'Variable defect not detected', with probabilities of $P_{VD}$ and $1 - P_{VD}$, respectively. Here, $P_{VD}$ represents the probability of detecting a variable defect in the source code.

In addition to the 'Variable' state, the model also has an 'Operation' state, which has two possible outcomes at the next level: 'Correct use of operation' and 'Incorrect use of operation', with probabilities of $1 - P_{def}$ and $P_{def}$, respectively.

The 'Incorrect use of operation' state then branches out into two possible outcomes at the next level: 'Operation defect detected' and 'Operation defect not detected', with probabilities of $P_{OD}$ and $1 - P_{OD}$, respectively. Here, $P_{OD}$ represents the probability of detecting an operation defect in the source code.

The conditional probability of a software defect can be defined as follows:

$$\eta = \frac{P_{\text{variable}DefectDetected} + P_{operationDefectDetected}}{P_{\text{variable}DefectDetected} + P_{\text{variable}DefectNotDetected} + P_{operationDefectDetected} + P_{operationDefectNotDetected}}$$

Here $P_{\text{variable}DefectDetected} = P_{\text{var}}P_{def}P_{VD}$, $P_{operationDefectDetected} = (1 - P_{\text{var}})P_{def}P_{OD}$
$P_{\text{variable}DefectNotDetected} = P_{\text{var}}P_{def}(1 - P_{VD})$, $P_{operationDefectNotDetected} = (1 - P_{\text{var}})P_{def}(1 - P_{OD})$

After substitution $P_{\text{variable}DefectDetected} \cdots \cdots P_{operationDefectDetected}$ in the source expression:

$$\eta = \frac{P_{\text{var}}P_{def}P_{VD} + (1 - P_{\text{var}})P_{def}P_{OD}}{P_{\text{var}}P_{def}P_{VD} + P_{\text{var}}P_{def}(1 - P_{VD}) + (1 - P_{\text{var}})P_{def}P_{OD} + (1 - P_{\text{var}})P_{def}(1 - P_{OD})}$$

$$\eta = P_{\text{var}}P_{VD} + (1 - P_{\text{var}})P_{OD} \tag{2}$$

As per Expression (2), the conditional probability of software defect detection depends on the probability of the software variables used in the source code and the conditional probabilities of detecting defects (defects of operations and defects of variables). We can determine the value of $P_{\text{var}}$ by analyzing the software code statically, i.e., without executing the code. However, to determine the values of $P_{VD}$ and $P_{OD}$, we would need to build additional software defect detection models.

### 2.4.2. Simple Model for Detection of Incorrect Use of Variables Based on Dimensional Analysis

Next, we introduce the simple model for the detection of incorrect use of variables based on dimensional analysis, as depicted in Figure 4.

This model has an initial state of 'Variable'. The initial state has two transitions to states 'OK' and 'Check Dimension', with probabilities $1 - P_{def}$ and $P_{def}$, respectively. In the state 'Check Dimension', we can evaluate the required physical dimension of the variable using dimensional analysis, such as length, mass, time, thermodynamic temperature, etc.

If the actual physical dimension is equal to the required physical dimension, we cannot detect the software defect. However, if they differ, we can identify the software defect. In

this case, the probabilities are $P_{\text{dim}}$ and $1 - P_{\text{dim}}$, where $P_{\text{dim}}$ represents the probability of two random variables having the same physical dimension.

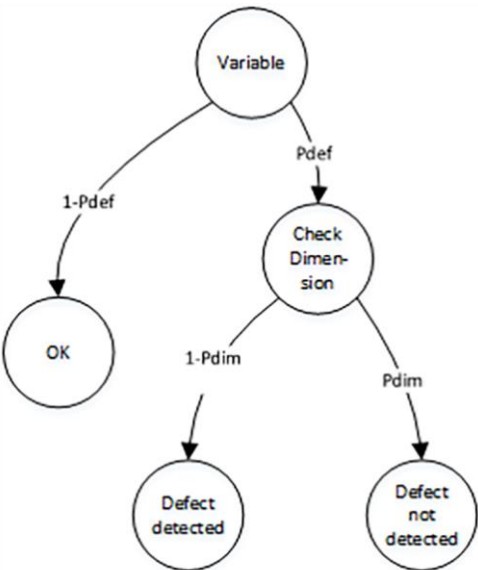

**Figure 4.** Simple model for the detection of incorrect use of variables based on dimensional analysis.

Let us define the conditional probability of defect detection of incorrect use of a program variable as follows:

$$P_{VD} = \frac{P_{defectDetected}}{P_{defectDetected} + P_{defectNotDetected}}.$$

Here $P_{defectDetected} = P_{def}(1 - P_{\text{dim}})$ and $P_{defectNotDetected} = P_{def}P_{\text{dim}}$

$$P_{VD} = 1 - P_{\text{dim}} \tag{3}$$

Let us consider a set of distinct software variables $\{\text{var}_1 \cdots \text{var}_{Nv}\}$ and a set of diverse physical dimensions $\{\text{dim}_1 \cdots \text{dim}_{Nd}\}$, where $N_V$ represents the cardinality of set $\{\text{var}_i\}$ and $N_D$ represents the cardinality of set $\{\text{dim}_j\}$.

To depict the relationship between these variables and dimensions, we can make use of an $N$-matrix (4):

$$
\begin{array}{c}
\\
\text{var}_1 \\
\text{var}_2 \\
\text{var}_3 \\
\text{var}_4 \\
\text{var}_5 \\
\text{var}_6 \\
\vdots \\
\text{var}_{N_V-1} \\
\text{var}_{N_V}
\end{array}
\begin{array}{ccccccccc}
\text{dim}_1 & \text{dim}_2 & \text{dim}_3 & \text{dim}_4 & \text{dim}_5 & \text{dim}_6 & \cdots & \text{dim}_{N_D-1} & \text{dim}_{N_D} \\
n_{11} & 0 & 0 & 0 & 0 & 0 & \cdots & 0 & 0 \\
n_{21} & 0 & 0 & 0 & 0 & 0 & \cdots & 0 & 0 \\
0 & n_{31} & 0 & 0 & 0 & 0 & \cdots & 0 & 0 \\
0 & 0 & n_{43} & 0 & 0 & 0 & \cdots & 0 & 0 \\
0 & 0 & 0 & n_{54} & 0 & 0 & \cdots & 0 & 0 \\
0 & 0 & 0 & n_{64} & 0 & 0 & \cdots & 0 & 0 \\
\vdots & \vdots & \vdots & \vdots & \vdots & \vdots & \ddots & \vdots & \vdots \\
0 & 0 & 0 & 0 & 0 & 0 & 0 & 0 & n_{N_V-1,N_D} \\
0 & 0 & 0 & 0 & 0 & 0 & 0 & 0 & n_{N_V,N_D}
\end{array}
\tag{4}
$$

The equation for the total quantity of usages of all software variables that have the same j dimension can be written as follows:

$$N_{VARj} = \sum_{i=1}^{N_V} n_{ij} \tag{5}$$

where $n_{ij}$ represents the total quantity usage of $i$-variable which has a $j$-physical dimension, and $N_V$ is the cardinality set of software variables.

Equation (5) shows the total number of variable usages in the code:

$$N_{VAR} = \sum_{i=1}^{N_V} \sum_{j=1}^{N_D} n_{ij} \tag{6}$$

To define the probability of choosing $i$-variable and $j$-variable with the same dimensions, we can use the total number of usages of variables with the $j$-physical dimension and the total number of usages of all variables in the code:

$$D_{ij} = \frac{n_{ij}}{N_{VAR}} \frac{\left(\sum_{i=1}^{N_V} n_{ij}\right) - n_{ij}}{N_{VAR} - n_{ij}} \tag{7}$$

According to (7), the probability of choosing two random variables that have the same physical dimension is given by the following equation:

$$P_{dim} = \sum_{i=1}^{N_V} \sum_{j=1}^{N_D} \left( \frac{n_{ij}}{N_{VAR}} \frac{\left(\sum_{k=1}^{N_V} n_{kj}\right) - n_{ij}}{N_{VAR} - n_{ij}} \right) \tag{8}$$

Here $n_{ij}$ represents the element of the $N$ matrix representing the quantity of usage for the $i$-variable with the $j$-physical dimension; $N_{VAR}$ represents the total number of variable usages in the code; $N_D$ represents the total number of different dimensions of variables in the code; and $N_V$ represents the total number of variables in the code.

For increasing the conditional probability detection of incorrect use of software variables we need to use other independent properties of variables. Using additional independent properties of variables can help increase the conditional probability detection of incorrect use of software variables. This is because using multiple properties helps to reduce the chance of false positives and increase the reliability of the detection model.

2.4.3. Simple Model for Detection of Incorrect Use of Variables Based on Orientational Analysis

In many cases, variables in CPS or the IoT have not only physical dimensions but also orientation information, which can be utilized to enhance the software quality of these systems. Therefore, we introduce a simple model for the detection of incorrect variable use based on orientational analysis (see Figure 5).

According to Figure 5, the initial state of the model is 'Variable'. This state has two transitions to states 'OK' and 'Check orientation' with probabilities $1 - P_{def}$ and $P_{def}$, respectively. In the state 'Check orientation', we can evaluate the required physical orientation of the variable using orientation analysis, such as $l_0$, $l_x$, $l_y$, and $l_z$. If the physical orientation of the variable matches the required orientation, we cannot detect a software defect. However, if the physical orientation is different from the required orientation, we can detect a software defect. These cases have probabilities $P_{orient}$ and $1 - P_{orient}$, where $P_{orient}$ is the probability that two randomly selected variables have the same physical orientation.

Now we need to define $P_{orient}$.

This defect model is similar to the dimension defect model of variables. However, in this case, we have four different orientations and $N_V$ different variables. We can describe the relationship between variables and their orientations using an $M$ matrix:

$$
\begin{array}{ccccc}
 & l_0 & l_x & l_y & l_z \\
\mathrm{var}_1 & m_{11} & 0 & 0 & 0 \\
\mathrm{var}_2 & m_{11} & 0 & 0 & 0 \\
\mathrm{var}_3 & 0 & m_{11} & 0 & 0 \\
\mathrm{var}_4 & 0 & m_{11} & 0 & 0 \\
\cdots & \cdots & \cdots & \cdots & \cdots \\
\mathrm{var}_{N_V} & 0 & 0 & 0 & m_{N_V,4}
\end{array}
\tag{9}
$$

where $l_k$ is a direction of orientation and $k = 0, x, y, z$.

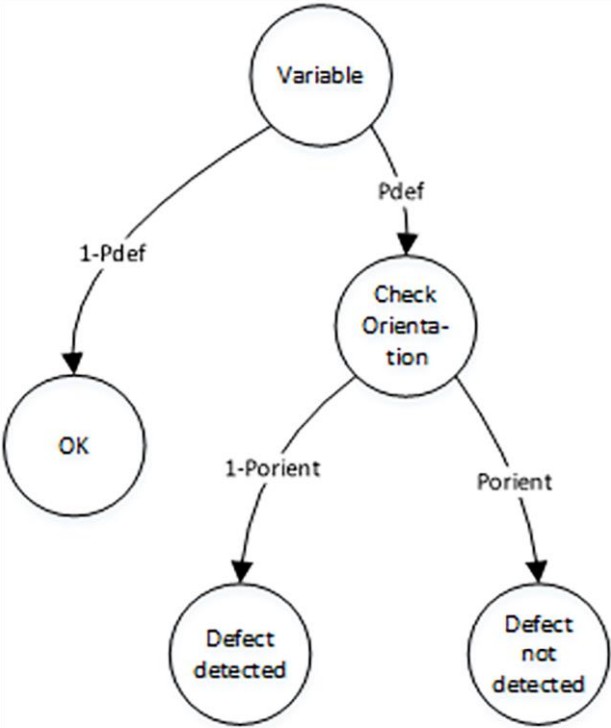

**Figure 5.** Simple model for the detection of incorrect use of variables based on orientational analysis.

Because, every software variable (as a host of a physical value) has only one orientation, every $i$-row of the $M$ matrix has only one non-zero number $m_{ik}$—the item of $M$ matrix—the number of using of $i$-variable which has $k$-orientation.

We can use Expression (8) for defining the probability of choosing two random variables that have the same physical orientation; this is given by the following Equation (10)

$$
P_{orient} = \sum_{i=1}^{N_V} \sum_{k=0,x,y,z} \left( \frac{n_{ik}}{N_{VAR}} \frac{\left( \sum_{n=1}^{N_V} m_{nj} \right) - m_{ik}}{N_{VAR} - m_{ik}} \right)
\tag{10}
$$

Here $m_{ik}$ represents the element of the $M$ matrix representing the quantity of usage for the $i$-variable with the $j$-physical orientation. $N_{VAR}$ represents the total number of variable usages in the code; $N_V$ represents the total number of variables in the code.

We can increase the conditional probability of software defect detection by concurrently using both dimensional and orientational analysis. By combining these two methods, we

can improve the accuracy of defect detection and reduce the likelihood of undetected defects.

### 2.4.4. Complex Model for Detection of Incorrect Use of Variables Based on Dimensional and Orientational Analysis

The complex model of the detection of incorrect use of variables based on both dimensional and orientational analysis is described in Figure 6.

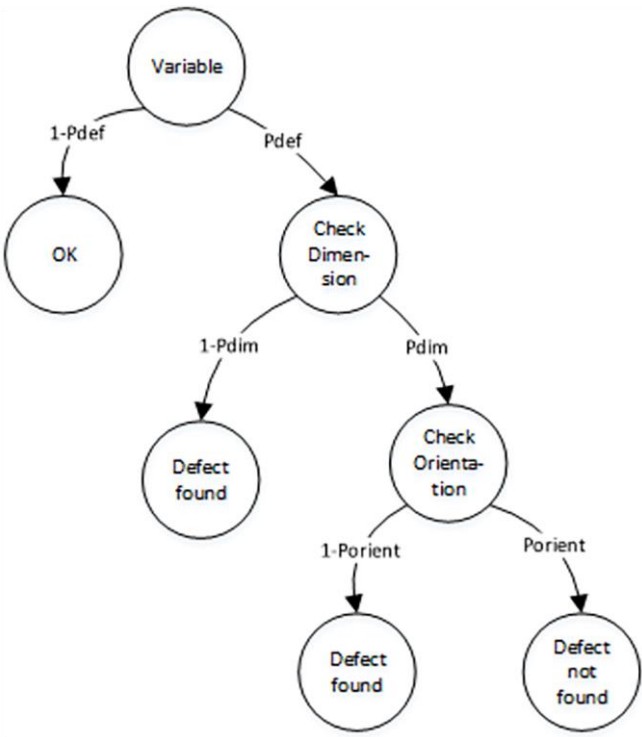

**Figure 6.** Complex model for the detection of incorrect use of variables based on dimensional and orientational analysis.

According to Figure 6, the initial state of the model is 'Variable'. This state has two transitions to the states 'OK' and 'Check Dimension' with probabilities $1 - P_{def}$ and $P_{def}$, respectively. In the 'Check Dimension' state, we evaluate the required physical dimension of the variable using dimensional analysis, such as length, mass, time, thermodynamic temperature, etc.

If the actual physical dimension matches the required dimension, we cannot detect the software defect with a probability of $P_{\dim}$. However, if they differ, we can identify the software defect with a probability of $1 - P_{\dim}$, where $P_{\dim}$ represents the probability that two randomly selected variables have the same physical dimension.

When we cannot detect the software defect, in the state 'Check orientation', we evaluate the required physical orientation of the variable using orientational analysis. If the physical orientation matches the required orientation, we cannot detect a software defect. However, if the physical orientation differs from the required orientation, we can detect a software defect. These cases have probabilities $P_{orient}$ and $1 - P_{orient}$, where $P_{orient}$ represents the probability that two randomly selected variables have the same physical orientation.

$$P_{VD} = \frac{P_{def}(1 - P_{\dim}) + P_{def}P_{\dim}(1 - P_{orient})}{P_{def}(1 - P_{\dim}) + P_{def}P_{\dim}(1 - P_{orient}) + P_{def}P_{\dim}P_{orient}}$$

$$P_{VD} = 1 - P_{\dim}P_{orient} \tag{11}$$

After the substitution of Expressions (8) and (10) in (11) we have

$$P_{VD} = 1 - \sum_{i=1}^{N_V} \sum_{j=1}^{4} \left( \frac{m_{ij}}{N_{VAR}} \frac{\left(\sum_{k=1}^{N_V} m_{kj}\right) - m_{ij}}{N_{VAR} - m_{ij}} \right) \sum_{i=1}^{N_V} \sum_{j=1}^{N_D} \left( \frac{n_{ij}}{N_{VAR}} \frac{\left(\sum_{k=1}^{N_V} n_{kj}\right) - n_{ij}}{N_{VAR} - n_{ij}} \right), \quad (12)$$

Here $m_{kj}$ represents the element of the $M$ matrix, which represents the quantity of usage for the $k$-variable with the $j$-physical orientation, $n_{ij}$ represents the element of the $N$ matrix representing the quantity of usage for the $i$-variable with the $j$-physical dimension, $N_D$ represents the total number of different dimensions of variables in the code, $N_V$ represents the total number of variables in the code, and $N_{VAR}$ represents the total number of variable usages in the code. Furthermore, there are four different orientations ($l_0$, $l_x$, $l_y$, $l_z$) in the code.

According to Equation (12), $P_{VD}$ denotes the conditional probability of detecting the incorrect use of software variables. The probability depends on the distribution of the software variables according to different dimensions and orientations.

To evaluate the correctness of the conditional probability of software defect detection, we need a model for detecting the incorrect use of operations. This model should take into account the types of operations that are commonly used in CPS and IoT software, as well as their potential incorrect use.

### 2.4.5. Model for Detection of Incorrect Use of Operations Based on Dimensional and Orientational Analysis

Let us consider three subsets of C++ operations: "additive" (A), "multiplicative" (M), and "other" (O) operations.

$$\begin{aligned}
&A = \{"+", "-", "=", "==", ">=", "<=", "!=", "<", ">", "++", "--", ".*", "->*", ",", ".", "->", "+=", "-=", "**"\}, \\
&M = \{"*", "/", "\%", "*=", "/=", "\%="\}, \\
&O = \{"||", "\&\&", "\&", "|", "^", "~", "<<", ">>", "::", "?", "<<=", ">>=", "\&=", "|=", "^="\}
\end{aligned} \quad (13)$$

In addition, we are given three probabilities associated with the utilization of this operation in the source code, namely, $P_A$, $P_M$, and $P_O$. Let us define the sum of these probabilities as the full group probability:

$$P_A + P_M + P_O = 1. \quad (14)$$

Let us define $P_A$, $P_M$, and $P_O$ as follows:

$$P_A = \frac{N_A}{N_A + N_M + N_O}, \quad P_M = \frac{N_M}{N_A + N_M + N_O}, \quad P_O = \frac{N_O}{N_A + N_M + N_O}, \quad (15)$$

Here, $N_A$ represents the total number of "additive" operations in a file, $N_M$ represents the total number of "multiplicative" operations in a file, and $N_O$ represents the total number of "other" operations in the file.

In this case, we can build a decision tree for the detection of incorrect use of operations based on dimensional and orientational analysis. The model allows us to define the conditional probability of operation defect detection (see Figure 7).

According to Figure 7, the initial state of the model is 'Operation'. This state has three transitions to the states 'Additive Operation', 'Multiplicative Operation', and 'Other Operation' with probabilities $P_A$, $P_M$, and $P_O$, respectively.

In the state 'Additive Operation', any 'additive' operation can be replaced by another operation. This event has the probability $P_{def}$. We then transition to the 'Using incorrect

operation 1' state with this probability. In the other case, with a probability of $1 - P_{def}$, we transition to the 'Using correct operation 1' state.

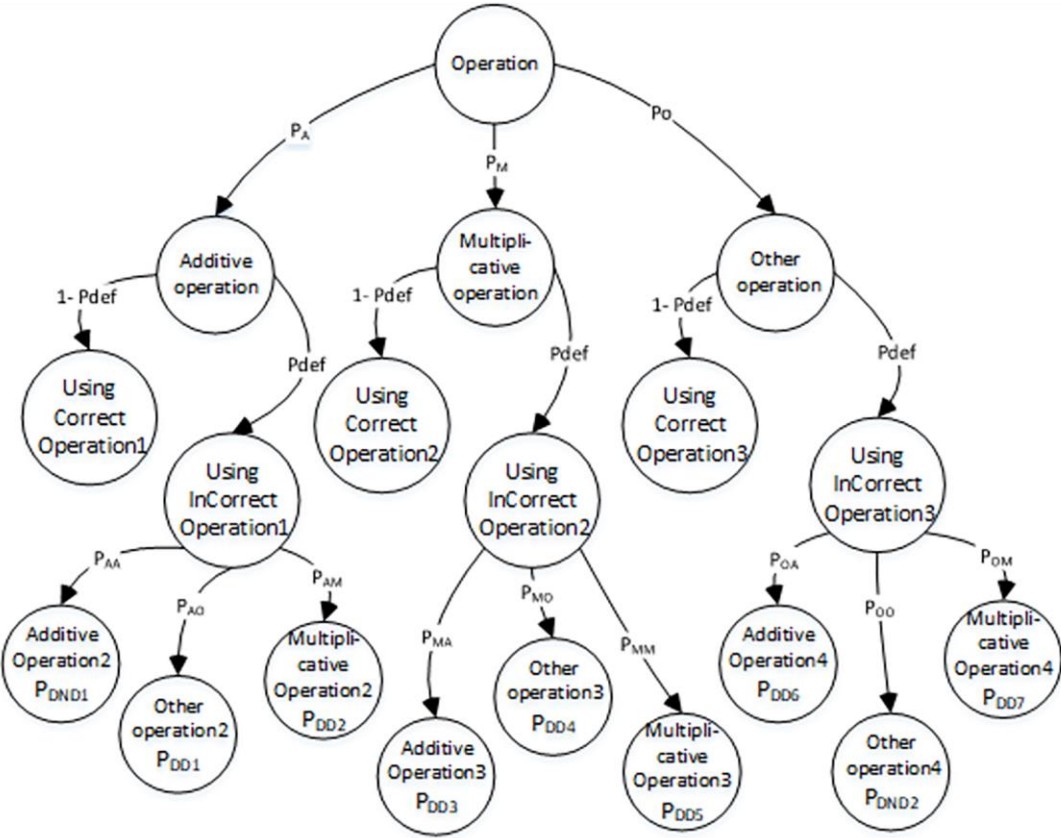

**Figure 7.** Model for the detection of incorrect use of operations based on dimensional and orientational analysis.

In the state 'Using incorrect operation 1', for a mutation where an additive operation mutates to another additive operation, we can transition to the 'Additive Operation 2' state with a probability of $P_{AA}$. In this case, we cannot detect a software defect, and the full probability for this scenario is $P_{DND1}$.

For a mutation where an additive operation mutates to other operations, we can transition to the 'Other Operation 2' state with a probability of $P_{AO}$. In this case, we can detect a software defect, and the full probability for this scenario is $P_{DD1}$.

For a mutation where an additive operation mutates to a multiplicative operation, we can transition to the 'Multiplicative Operation 2' state with a probability of $P_{AM}$. In this case, we can detect a software defect, and the full probability for this scenario is $P_{DD2}$.

In the state 'Multiplicative Operation', any 'multiplicative' operation can be replaced by another operation. This event has the probability $P_{def}$, and we transition to the 'Using incorrect operation 2' state. In the other case, with a probability of $1 - P_{def}$, we transition to the 'Using correct operation 2' state.

In the state 'Using incorrect operation 2', for a mutation where a multiplicative operation mutates to other additive operations, we can transition to the 'Additive Operation 3' state with a probability of $P_{MA}$. In this case, we can detect a software defect. The full probability for this scenario is $P_{DD3}$.

For a mutation where a multiplicative operation mutates to 'other' operations, we can transition to the 'Other Operation 3' state with a probability of $P_{MO}$. In this case, we can detect a software defect. The full probability for this scenario is $P_{DD4}$.

For a mutation where a multiplicative operation mutates to other multiplicative operations, we can transition to the 'Multiplicative Operation 3' state with a probability of

$P_{MM}$. In this case, we can detect a software defect. The full probability for this scenario is $P_{DD5}$.

In the state 'Other Operation', any 'other' operation can be replaced by another operation. This event has the probability $P_{def}$, and we transition to the 'Using incorrect operation 3' state. In the other case, with a probability of $1 - P_{def}$, we transition to the 'Using correct operation 3' state.

In the state 'Using incorrect operation 3', for a mutation where an 'other' operation mutates to additive operations, we can transition to the 'Additive Operation 4' state with a probability of $P_{OA}$. In this case, we can detect a software defect. The full probability for this scenario is $P_{DD6}$.

For a mutation where an 'other' operation mutates to other 'other' operations, we can transition to the 'Other Operation 4' state with a probability of $P_{MO}$. In this case, we cannot detect a software defect. The full probability for this scenario is $P_{DND2}$.

For a mutation where an 'other' operation mutates to multiplicative operations, we can transition to the 'Multiplicative Operation 4' state with a probability of $P_{OM}$. In this case, we can detect a software defect. The full probability for this scenario is $P_{DD7}$.

Additionally, we can conclude, that $P_{AA} + P_{AM} + P_{AO} = 1$, $P_{MA} + P_{MM} + P_{MO} = 1$, $P_{OA} + P_{OM} + P_{OO} = 1$. We can define the conditional probability of detecting incorrect use of software operations as follows:

$$P_{OD} = \frac{\sum\limits_{i=1}^{7} P_{DDi}}{\sum\limits_{i=1}^{7} P_{DDi} + P_{DND1} + P_{DND2}}.$$

Here, $P_{DD1} = P_A P_{def} P_{AO}$, $P_{DD2} = P_A P_{def} P_{AM}$, $P_{DND1} = P_A P_{def} P_{AA}$, $P_{DD3} = P_M P_{def} P_{MA}$, $P_{DD4} = P_M P_{def} P_{MO}$, $P_{DD5} = P_M P_{def} P_{MM}$, $P_{DD6} = P_O P_{def} P_{OA}$, $P_{DD7} = P_O P_{def} P_{OM}$, $P_{DND2} = P_O P_{def} P_{OO}$.

After substitution

$$P_{OD} = \frac{P_A P_{AO} + P_A P_{AM} + P_M + P_O P_{OA} + P_O P_{OM}}{P_A + P_M + P_O}$$

According to $P_A + P_M + P_O = 1$, $P_{OD} = P_A(1 - P_{AA}) + P_M + P_O(1 - P_{OO})$. Because $P_{AA} \approx P_A$ and $P_{OO} \approx P_O$, then $P_{OD} \approx P_A(1 - P_A) + P_M + P_O(1 - P_O)$

$$P_{OD} \approx 1 - P_A^2 - P_O^2 \tag{16}$$

Here, $P_A$ represents the conditional probability of 'additive' operations in a file and $P_O$ represents the conditional probability of 'other' operations in a file.

According to Equation (16), $P_{OD}$ denotes the conditional probability of detecting the incorrect use of software operations. The value of $P_{OD}$ depends on the square of the probabilities of additional operations (such as $+$, $-$, $=$, $<$, etc.) and other operations (such as = {"||", "&&", "&", "|", "^", "~", "<<", ">>", "::", "?" etc.) in a source file. In order to evaluate $P_{OD}$, it is necessary to define the values of $P_A$ and $P_O$. This evaluation requires analyzing the real source code.

Using Equation (2), which defines the conditional probability of software defect detection as a function of $P_{var}$ (defined by the source code of the file) and the conditional probabilities of variable usage defect detection ($P_{VD}$, defined by Equation (12)) and operation usage defect detection ($P_{OD}$, defined by Equation (16)), allows us to evaluate the conditional probability of software defect detection.

## 3. Results

After analyzing the source code of Unmanned Aerial Vehicle Systems, which had a total volume of 2 GB and a total number of files of 20,000, saved on GitHub using our own

statistical analyzer, we were able to determine the necessary statistical characteristics of the C++ source code.

Distribution $N_{var}$—total number of different variables per file (Figures 8 and 9).

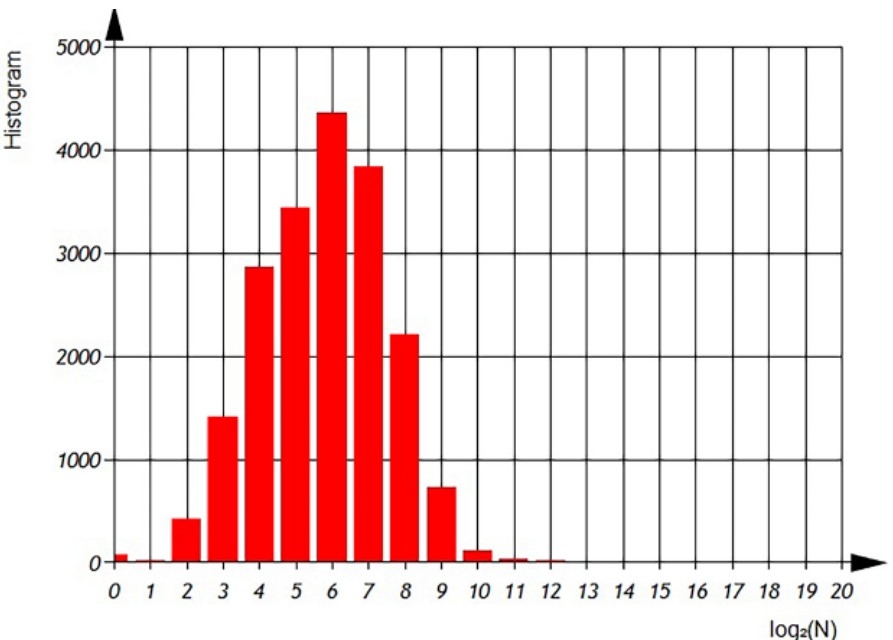

**Figure 8.** Histogram of the total number of variables per file (semi-log scale).

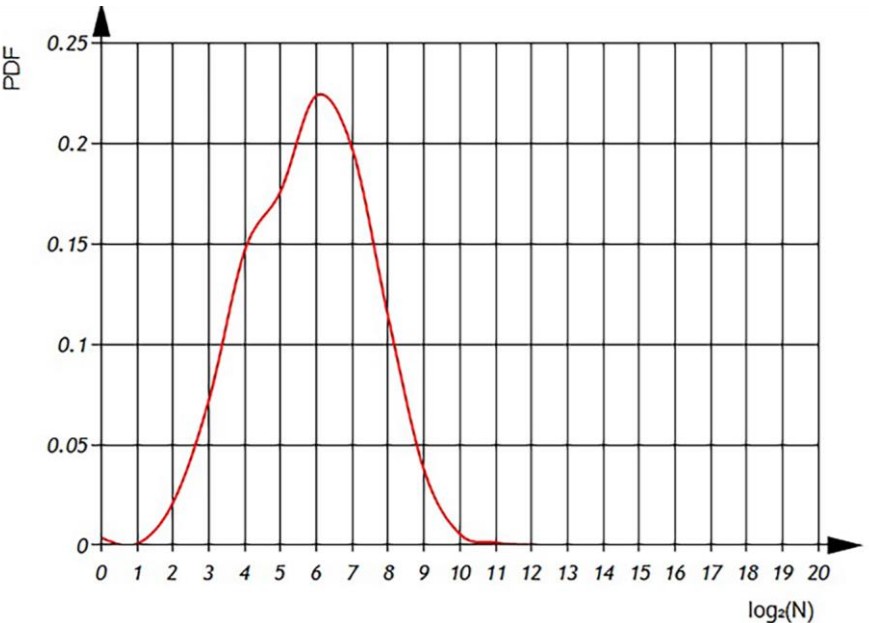

**Figure 9.** Probability density function of total variables per file (semi-log scale).

According to Figures 8 and 9 we can observe the distribution of different variables per file in semi-logarithmic coordinates. The histogram (Figure 8) reveals that the average number of different variables is $2^6$, with a maximum of 64 variables observed in 4500 files. However, there are files that contain only one variable, as well as files with 1024 variables. The sum of the histogram columns corresponds to the total number of files, which is 20,000.

In Figure 9, presented subsequently, we can examine the probability density function of variable distributions. It is important to note that the integral of the probability density function should always equal one, ensuring a proper probability distribution.

According to Figures 10 and 11 we can observe the distribution of $N_{VF}$, which represents the average number of variables uses per file, in semi-logarithmic coordinates. The histogram (Figure 10) shows that the average number of variable usages is 5. The sum of the histogram columns corresponds to the total number of files, which is 20,000.

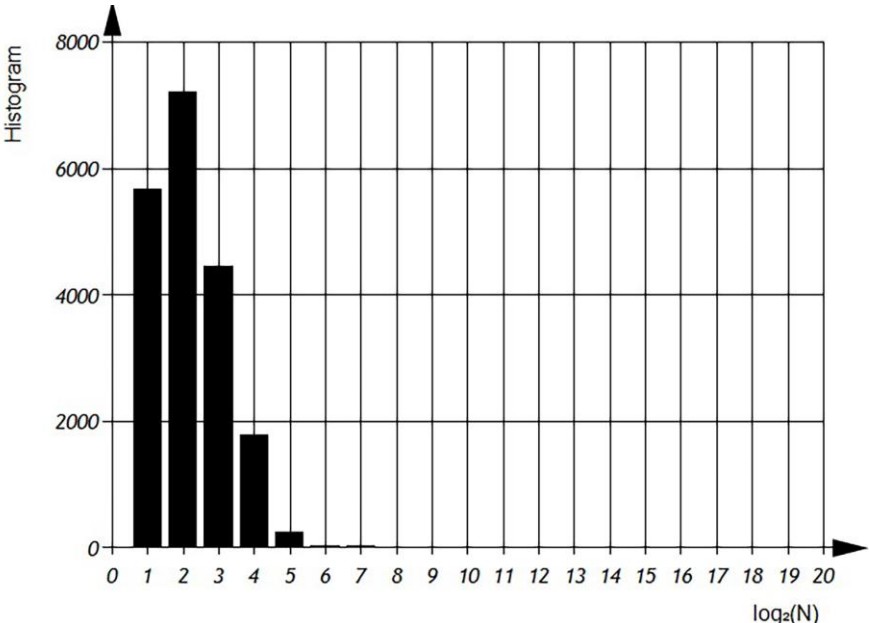

**Figure 10.** Histogram of variable usage per file using a semi-logarithmic scale.

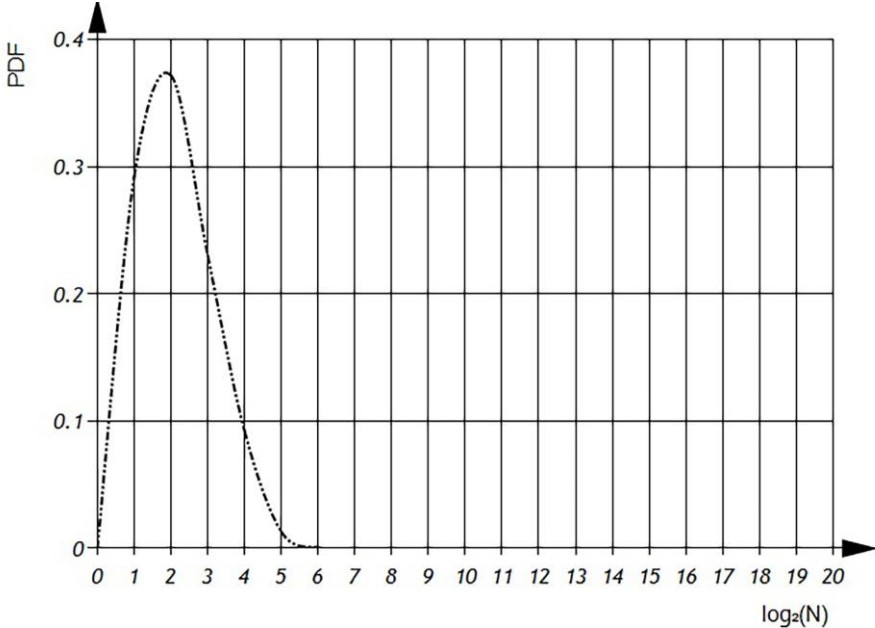

**Figure 11.** Probability density function of variable usage per file (semi-log scale).

In the subsequently presented Figure 11, we can analyze the probability density function of the distributions for the average usage of variables. According to the Probability Density Function, we can see that there are files with an average usage of 64 variables. It is crucial to note that the integral of the probability density function should always equal one, ensuring a proper probability distribution.

According to Equation (12) and the distributions described in Figures 8 and 10, as well as the uniform distribution for both $n_{ij}$ and $m_{ij}$, we can obtain the Probability Density

Functions of conditional probability of detecting incorrect use of software variables $P_{VD}$, shown in Figure 12.

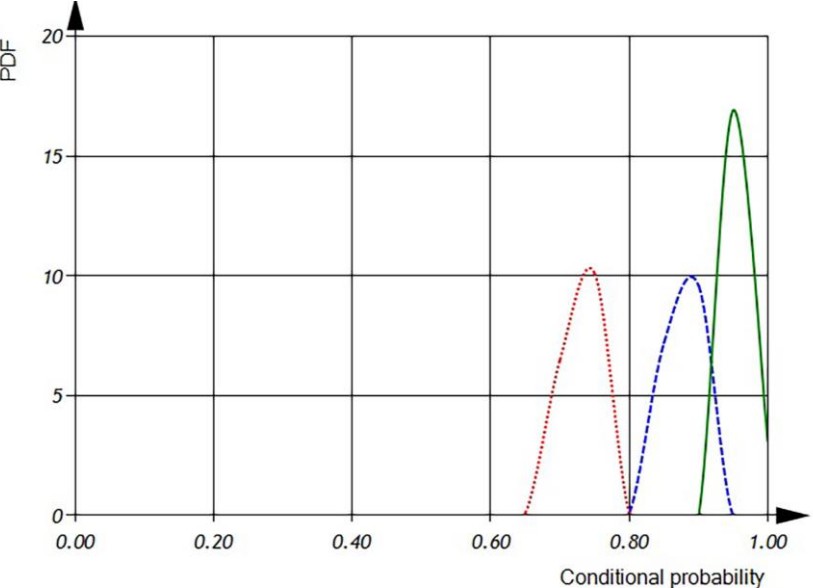

**Figure 12.** Probability density functions of conditional probability for detecting the incorrect use of software variables: (1) dimensional analysis (blue dash line); (2) orientational analysis (red dot line); (3) orientational and dimensional analysis (green solid line).

According to Figure 12, when combining dimensional and orientational analysis, the conditional probability for detecting incorrect use of software variables is greater than 0.9.

Additional statistical characters of C++ source code was evaluated: $N_A$—total numbers of additive operations per file; $N_O$—total numbers of 'other' operations per file; $N_M$—total numbers of multiplicative operations per file. Corresponded distributions shown on the Figures 13 and 14.

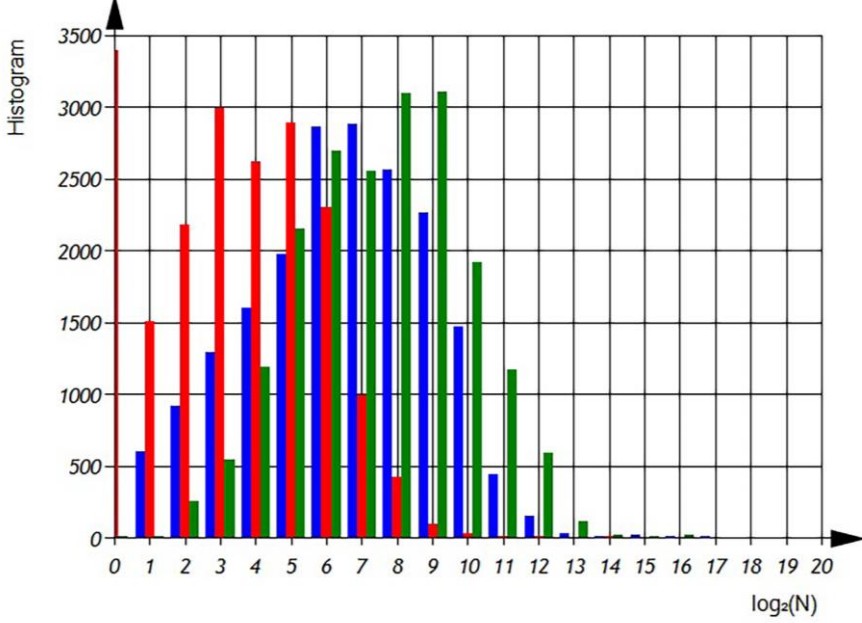

**Figure 13.** Histogram of software operations for the total number of additive operations (blue lines), multiplicative operations (red lines), and other operations (green lines) per file on semi-logarithmic coordinates.

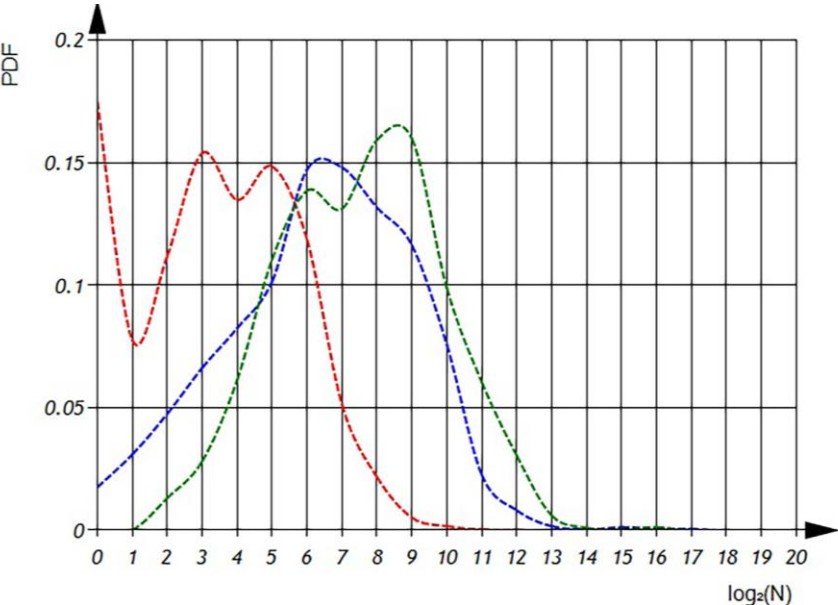

**Figure 14.** Probability density functions of software operations for the total number of additive operations (blue line), multiplicative operations (red line), and other operations (green line) per file on semi-logarithmic coordinates.

In Figure 14 we can observe the distributions of $N_A$, $N_M$, and $N_O$, which represent the average number of operations per file in semi-logarithmic coordinates. The histograms show that the average number of 'additive' operations is 128, observed in 2500 files. However, there are files that only contain 2 additive operations, as well as files with an average usage of 8192 additive operations. The sum of the histogram columns corresponds to the total number of files, which is 20,000. The histogram shows that the average number of 'multiplicative' operations is 16, observed in 2600 files. However, there are files that only contain 2 multiplicative operations, as well as files with an average usage of 512 multiplicative operations. The histogram shows that the average number of 'other' operations is 256, observed in 3000 files. However, there are files that only contain 4 'other' operations, as well as files with an average usage of 4096 'other' operations. The sum of the histogram columns corresponds to the total number of files, which is 20,000.

In the subsequently presented, we can examine the probability density function of the distributions for different operations per file. It is crucial to note that the integral of the probability density function should always equal one, ensuring a proper probability distribution.

By referring to Equation (15) and the distributions of $N_A$ (total number of additive operations), $N_M$ (total number of multiplicative operations), and $N_O$ (total number of other operations) (as shown in Figure 15), we can calculate the distributions of $P_A$, $P_M$, and $P_O$.

In Figures 15 and 16, we can observe the distribution of conditional probabilities of operations per file in semi-logarithmic coordinates. The histogram shows that the average of conditional probabilities of operations per file. The value of conditional probability of 'additive' operations is $P_A = 0.309 \pm 0.161$ [0.000 ... 0.75]. The value of conditional probability of 'multiplicative' operations is $P_M = 0.056 \pm 0.056$ [0.000 ... 0.636]. The value of conditional probability of 'other' operations is $P_O = 0.635 \pm 0.155$ [0.2 ... 0.992].

In the subsequently presented, we can examine the probability density functions of the conditional probability of different operations. It is crucial to note that the integral of the probability density function should always equal one, ensuring a proper probability distribution.

Based on the distributions of $P_A$, $P_M$, and $P_O$ we can calculate the distribution of the conditional probability of operation defect detection $P_{OD}$ (as depicted in Figure 17 The

histogram reveals that the average conditional probability of operation defect detection per file is $P_{OD} = 0.45 \pm 0.161$ [0.000 ... 0.8].

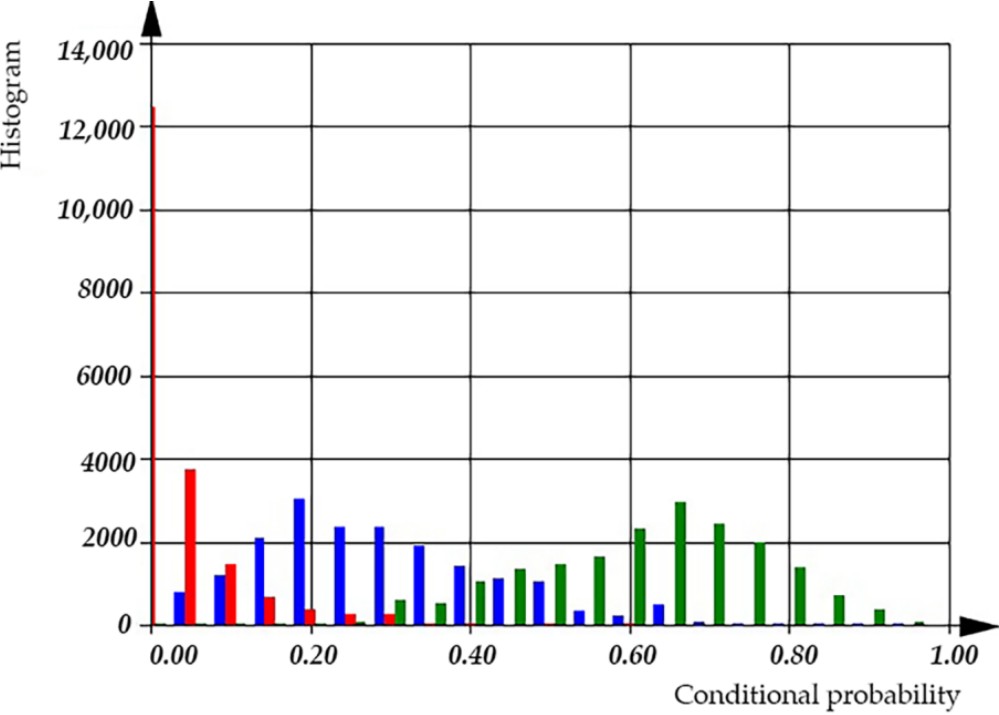

**Figure 15.** Histogram of the conditional probability of additive operations (blue lines), multiplicative operations (red line), and other operations (green line) per file based on dimensional and orientational analysis.

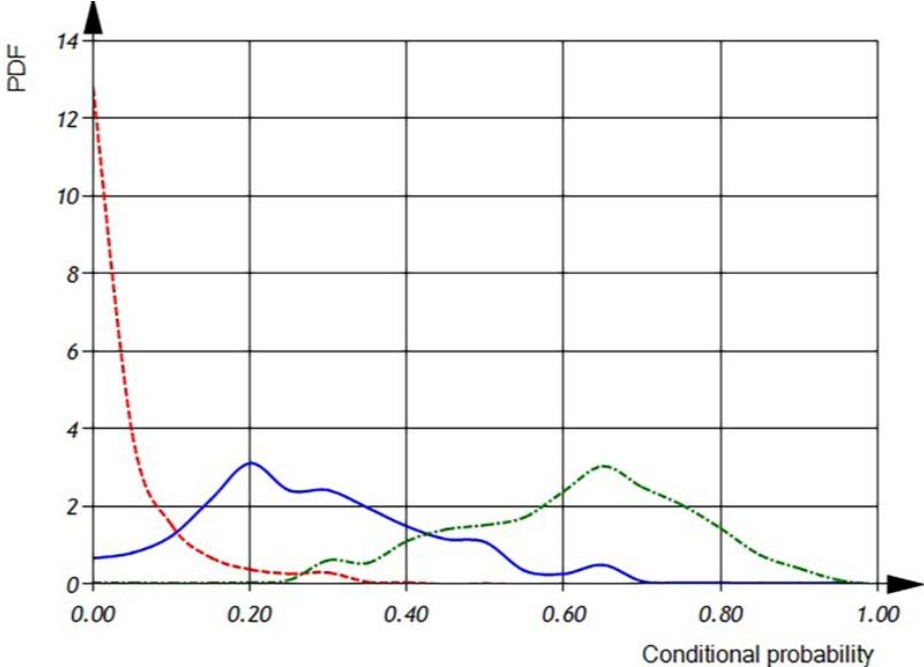

**Figure 16.** Probability density functions of the conditional probability of additive operations (blue solid line), multiplicative operations (red dash line), and other operations (green dash-dot line) per file based on dimensional and orientational analysis.

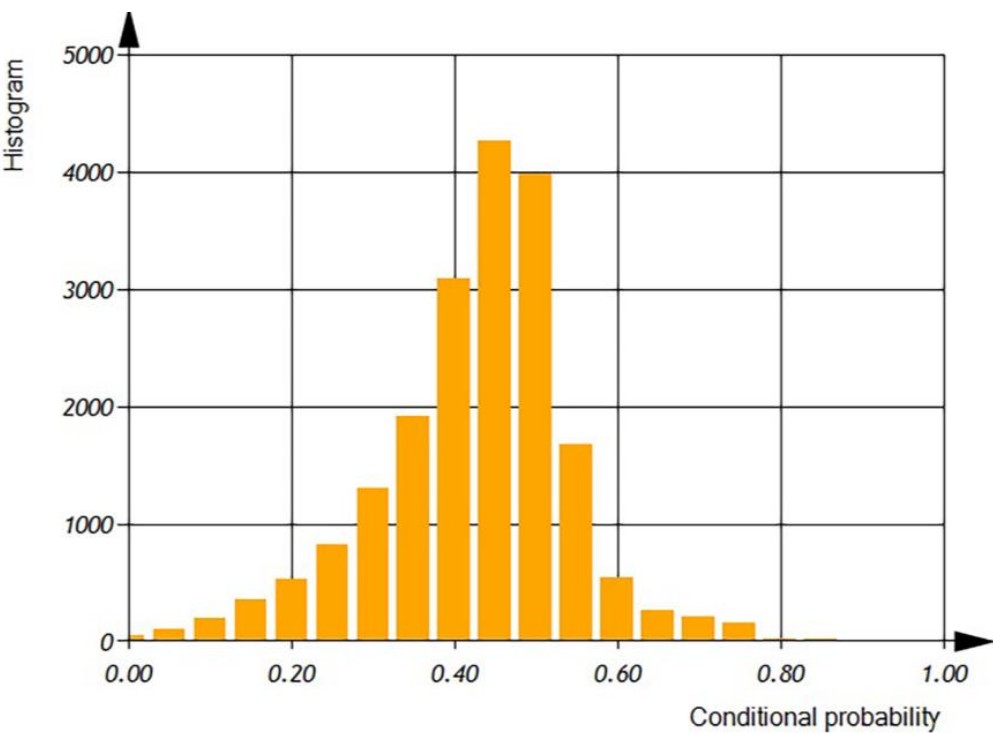

**Figure 17.** Histogram of the conditional probability of software operation defect detection based on dimensional and orientational analysis.

According to Figure 18, we can see that the mean value of the conditional probability of software operation defect detection is 0.5.

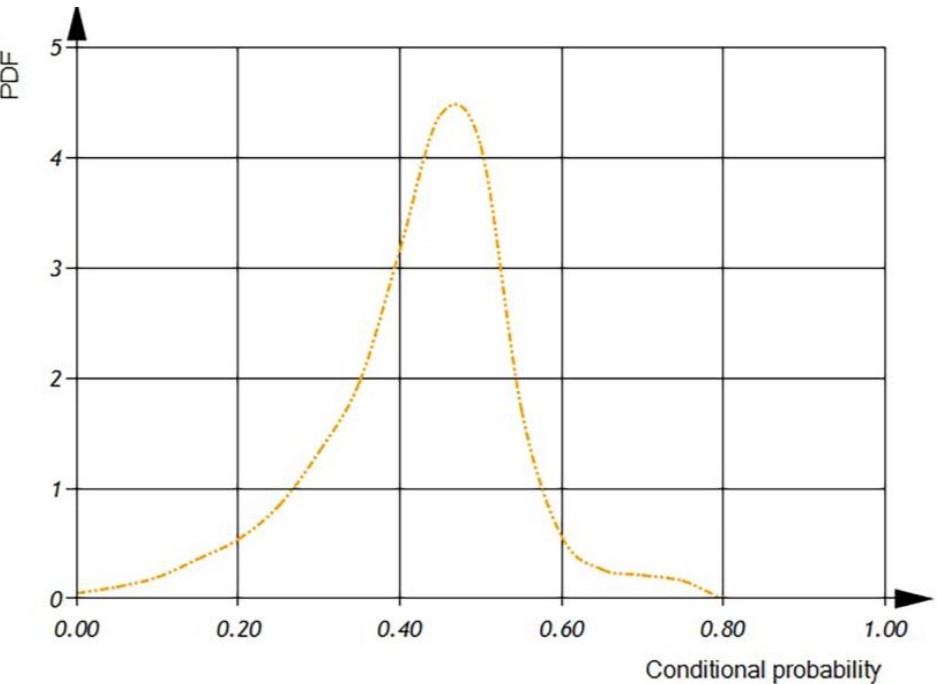

**Figure 18.** Probability density function of the conditional probability of software operation defect detection based on dimensional and orientational analysis.

After analyzing real C++ code statistically, we built the distributions of $P_O$ and $P_{var}$ (as shown in Figures 19 and 20).

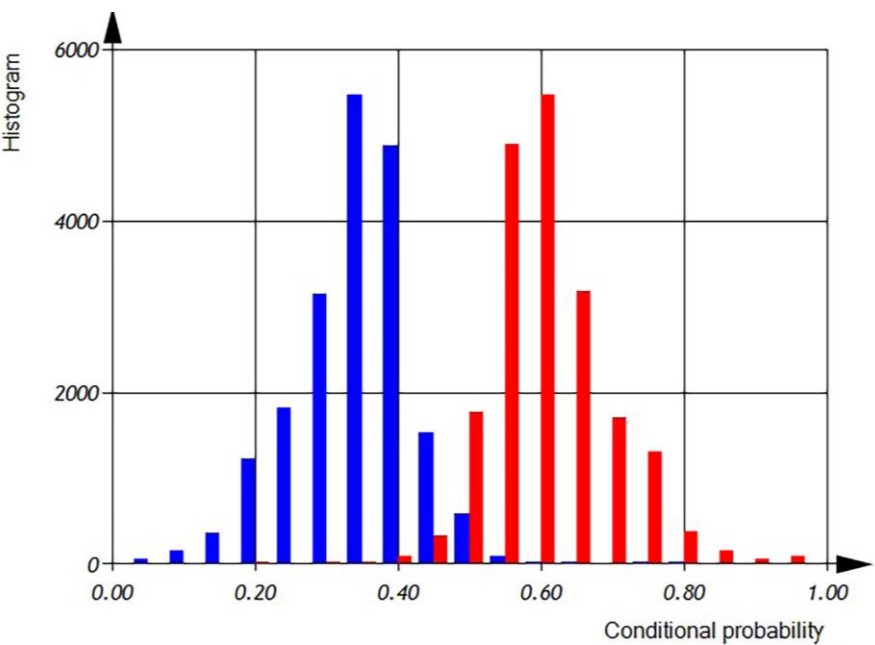

**Figure 19.** Histogram of conditional probabilities of variables ($P_{var}$—blue lines) and operations ($P_O$—red lines) per file.

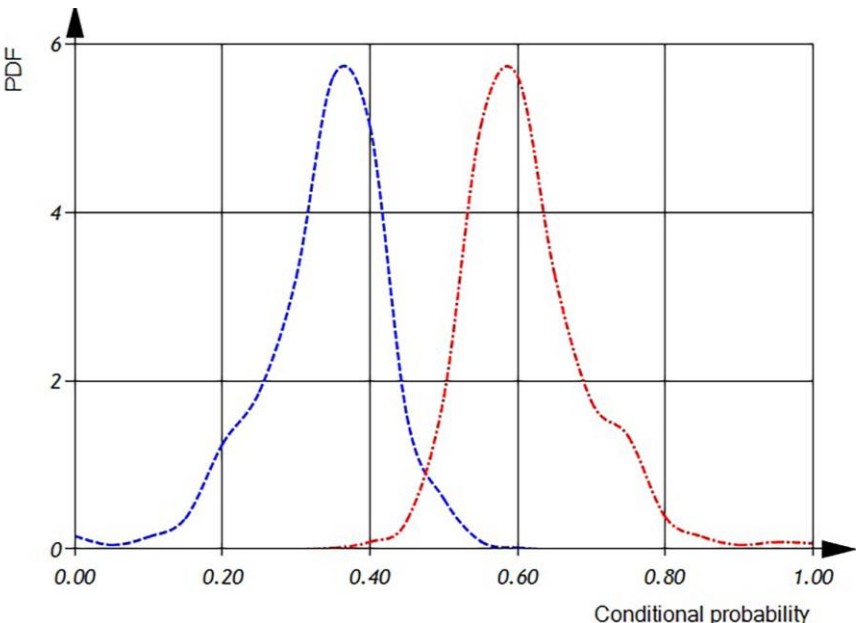

**Figure 20.** Probability density functions of conditional probabilities of variables ($P_{var}$—blue dash line) and Operations ($P_O$—red dash-dot line) per file.

The histogram shows that the average conditional probability of operations is $P_O = 0.65 \pm 0.12 \ [0.4 \dots 0.9]$. Similarly, the histogram reveals that the average conditional probability of variables is $P_{var} = 0.35 \pm 0.12 \ [0.000 \dots 0.55]$.

Figure 20, displays two distributions of the conditional probability for software operations and software variables, with peaks at 0.35 and 0.65, satisfying the equation $P_{var} + P_O = 1$.

Now, we can calculate the conditional probability of software defect detection based on the embedded source code and the proposed defect models (see Expression (2)). Let $\eta$ be the conditional probability, then we have $\eta = P_{var}P_{VD} + (1 - P_{var})P_{OD}$, where $P_{var}$ is the conditional probability of variables in the source code, $P_{VD}$—is the conditional probability

of the defects detection of variables using the defect in the source code, and $P_{OD}$ is the conditional probability the defects detection of operation using defect in the source code.

In the upcoming figures, Figure 21 presents a histogram of the conditional probabilities of defect detection, while Figure 22 shows probability density functions of software defect detection.

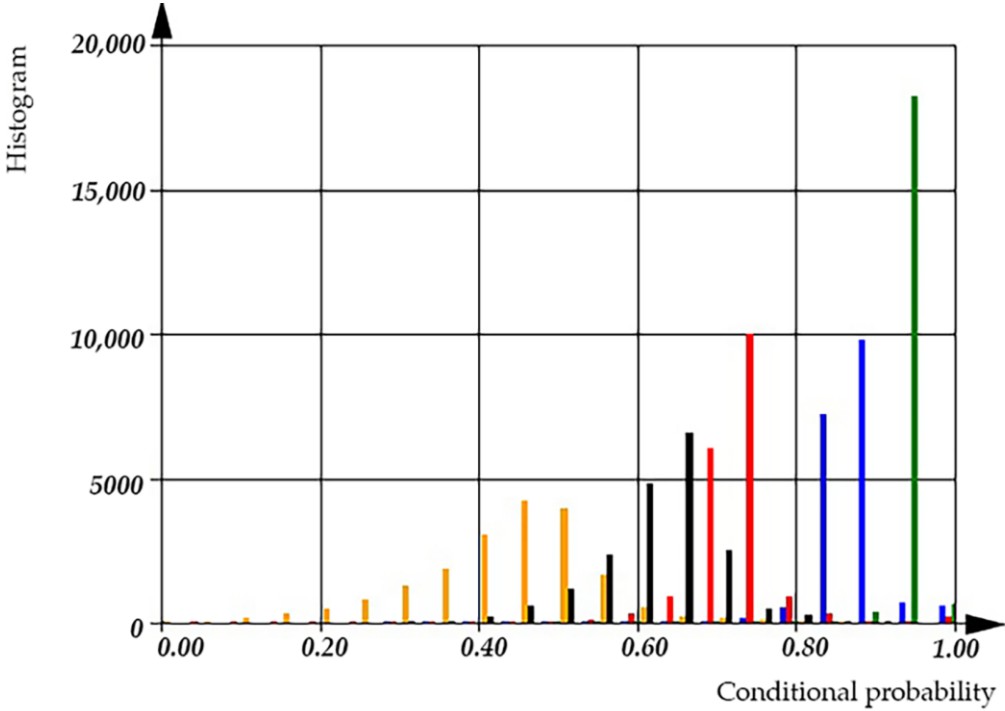

**Figure 21.** Histogram of conditional probabilities for the detection of (1) incorrect use of software operations and variables based on dimensional and orientational analysis (black solid lines); (2) incorrect use of software operations (orange lines); (3) incorrect use of software variables based on orientational analysis (red lines); (4) incorrect use of software variables based on dimensional analysis (blue lines); (5) incorrect use of software variables based on dimensional and orientational analysis (green lines).

According to the histogram (see Figure 21), the conditional probability for the detection of incorrect usage of variables is 0.95, while the conditional probability for the detection of incorrect usage of operations has a mean of 0.5. We can observe that the conditional probability of the incorrect usage of variables increases after incorporating both dimensional and orientational analysis. Using dimensional analysis alone yields a conditional probability of a defect detection of 0.9, while orientational analysis alone provides a conditional probability of a defect detection of 0.73. However, the overall conditional probability for the detection of the incorrect usage of operations or variables has a mean value of 0.60.

According to Figure 22, the conditional probability for the detection of the incorrect usage of variables has a mean value of 0.95 and is distributed in the interval of 0.4 to 0.8, while the conditional probability for the detection of the incorrect usage of operations has a mean of 0.5 and is distributed in the interval of 0 to 0.65. It is evident that the conditional probability of the incorrect usage of variables increases after incorporating both dimensional and orientational analysis. The incorporation of both analysis methods narrows the interval of distribution. When using dimensional analysis alone, the conditional probability of a defect detection is 0.9 within the interval of 0.8 to 0.95, while orientational analysis alone provides a conditional probability of a defect detection of 0.73 within the interval of 0.6 to 0.83. However, the overall conditional probability for the detection of the incorrect usage of operations or variables has a mean value of 0.60 within the interval of 0.4 to 0.75.

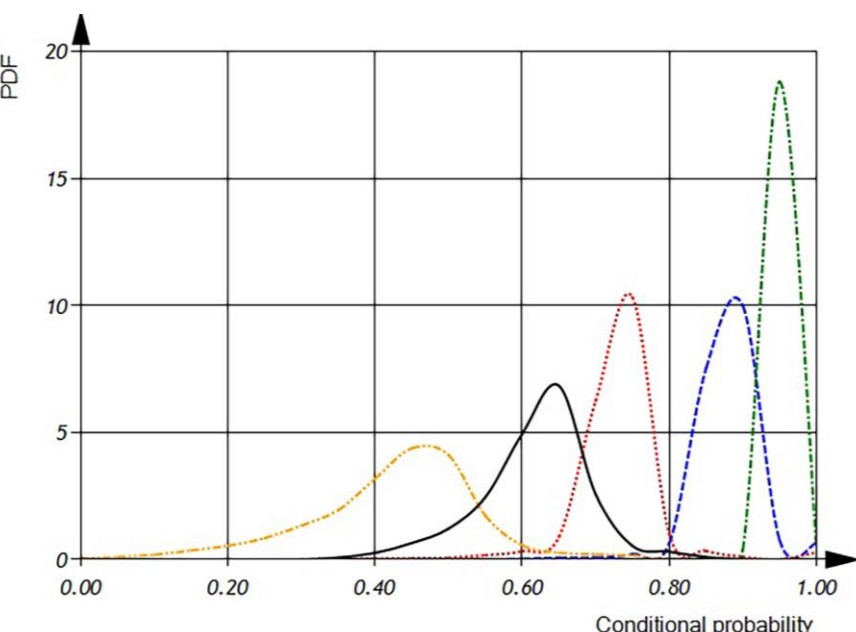

**Figure 22.** Probability density function of conditional probabilities for the detection of (1) incorrect use of software operations and variables based on dimensional and orientational analysis (black solid line); (2) incorrect use of software operations (orange dash-dot line); (3) incorrect use of software variables based on orientational analysis (red dot line); (4) incorrect use of software variables based on dimensional analysis (blue dash line); and (5) incorrect use of software variables based on dimensional and orientational analysis (green dash-dot line).

## 4. Discussion

The proposed method of formal software verification based on dimensional analysis and orientational analysis appears to be an effective approach to detecting software defects. The fact that it can detect over 60% of software defects, including those related to the incorrect usage of variables, operations, and functions, is noteworthy and suggests that it could be a valuable tool in software development.

However, it is important to note that no single method can detect all types of software defects, and different methods may be better suited for different types of defects. Thus, while the proposed method shows promise, it should be evaluated and compared to other approaches to determine its overall efficacy and limitations.

To fully assess its overall effectiveness and limitations, it is crucial to develop a concrete tool that can evaluate and compare the proposed method with other existing tools.

In the future, the proposed method will demonstrate a high level of effectiveness, detecting 90% of incorrect uses of software variables and more than 50% of incorrect uses of operations. One of the imminent tasks is to create a type of library for the formal verification of CPS and IoT software during compile-time.

## 5. Conclusions

This article focuses on a formal software verification method based on software invariants derived from both dimensional and orientational analysis.

The advantages of the proposed method are as follows:

1. Early detection of software defects in compile-time.
2. Reduced testing time via formal verification in compile-time and run-time
3. By catching a large number of defects early on, the method can help minimize the need for extensive debugging, maintenance, or post-release updates, resulting in overall cost reduction.
4. The ability to detect over 60% of latent defects suggests that the method contributes to improving software quality.

5.  The method can be seen as a complementary approach that focuses on detecting latent defects based on software characteristics.
6.  Continuous improvement efforts can contribute to even higher detection rates, leading to further advancements in software quality assurance.

Overall, the high detection rate of the proposed method and its potential benefits in reducing testing time and improving both reliability and software quality demonstrate its value as an efficient and effective approach to software defect detection.

However, the method has certain limitations, such as the need to know the physical dimensions and orientations of source variables at compile-time. Nonetheless, it offers several advantages, including improved programmer productivity, as programmers no longer need to spend time tracking down dimensional and orientational errors during development and run-time. Additionally, the method enables a comprehensive analysis of dimensional and orientational correctness during compile-time and run-time, including the correct use of software variables, operations, functions, and procedures through added argument checking.

Although the proposed method has the potential to enhance software reliability, it requires further research and development of specialized analysis tools to realize its full effectiveness.

**Author Contributions:** Conceptualization, Y.M.; methodology, Y.M. and Y.S.; software, Y.M. and Y.S.; validation, Y.M. and Y.S.; formal analysis, Y.M. and Y.S.; resources, Y.M. and Y.S.; data curation, Y.M.; writing—original draft preparation, Y.M. and Y.S.; writing—review and editing, Y.M. and Y.S.; visualization, Y.M. and Y.S.; supervision, Y.M.; project administration, Y.S. All authors have read and agreed to the published version of the manuscript.

**Funding:** This research received no external funding.

**Data Availability Statement:** Data are contained within this article.

**Conflicts of Interest:** The authors declare no conflict of interest.

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
