# Peer review of "A Software Verification Method for the Internet of Things and Cyber-Physical Systems"

_computation, doi:10.3390/computation11070135_

Round 1

Reviewer 1 Report

The core of the manuscript,  A Software Verification Method for Internet of Things and Cyber-Physical Systems is an interesting topic for a broad audience, as it would trigger research from different areas. The paper concept is impressive, and this research work’s novelty is significant. My feedback on this paper is given below:

Can you explain in more detail how the physical quantities defined by the System International and their homogeneity in software code form the basis of your software verification method? How does this approach contribute to ensuring functional and high-quality software for IoT and Cyber-Physical Systems?

You mentioned that the proposed method cannot check expressions with angles, angle speed, and similar features. Could you elaborate on the transformation for physical value orientation introduced by Siano and how it enables the verification of software code in these cases? What are the key considerations and challenges in incorporating orientational homogeneity into the software verification process?

How did you develop the special software defect models based on the statistical characteristics of software code? Could you provide some insights into the statistical analysis tool used and the specific characteristics analyzed in the 2 GB of C++ GITHUB code for drones? What were the main findings or observations from this analysis?

In your evaluation of the proposed method, you mentioned that it can detect over 60% of latent software defects based on the actual distribution of software characteristics. Can you discuss the implications of this detection rate and how it compares to existing software verification methods? What are the potential benefits for reducing testing time, improving reliability, and enhancing software quality?

How scalable is the proposed software verification method? Does it have any limitations or considerations when applied to larger software systems or more complex IoT and Cyber-Physical Systems? Are there any specific factors or characteristics that may impact the effectiveness of the method in different contexts?

Can you provide some examples or scenarios where the proposed software verification method has been applied or could be applied in practice? How does it fit into the overall software development lifecycle for IoT and Cyber-Physical Systems? Are there any specific industries or domains where this method could have significant impact or benefits?

Based on your research, what are the key insights or recommendations for software developers or organizations looking to improve the verification of IoT and Cyber-Physical Systems software? Are there any specific best practices or guidelines that you would suggest based on the findings of your study?

Are there any potential future developments or extensions to the proposed method that you are currently exploring or would like to investigate? Are there any open research questions or areas where further research is needed to enhance or refine the software verification process for IoT and Cyber-Physical Systems?

How does your software verification method integrate with other existing software engineering practices, such as testing, code review, or formal methods? Can you discuss the complementarity or synergies between your approach and these established techniques?

Author can read the following papers to improve the quality of research:10.1109/JAS.2021.1004003, 10.1016/j.jpdc.2021.03.011, 10.3390/electronics11193070

What are the main contributions or novel aspects of your research in the context of software verification for IoT and Cyber-Physical Systems? How does your work advance the current state of the art and contribute to the broader field of software engineering and system reliability?

Moderate editing of English language needed

Author Response

Response to Reviewer 1 Comments

Point 1: Can you explain in more detail how the physical quantities defined by the System International and their homogeneity in software code form the basis of your software verification method? How does this approach contribute to ensuring functional and high-quality software for IoT and Cyber-Physical Systems?

Response 1: Certainly! The System International (SI) is a globally recognized and widely used system of units for measuring physical quantities. It defines a set of base units, such as meters for length, kilograms for mass, and seconds for time, along with derived units, which are combinations of base units, such as meters per second for velocity or kilograms per cubic meter for density.

When it comes to software code, developers often need to work with and manipulate physical quantities in their programs. To ensure the correctness of such code, formal software verification methods can be applied. These methods use mathematical techniques to formally prove properties about the software, such as its correctness, safety, or absence of certain errors.

The concept of homogeneity, derived from the SI system, plays a significant role in formal software verification. Homogeneity refers to the idea that equations involving physical quantities must be dimensionally consistent. In other words, the units on both sides of an equation must match.

When developing software code that involves physical quantities, developers can utilize static type systems or programming languages with static type checking to enforce homogeneity. By assigning appropriate types to variables and functions representing physical quantities, the compiler or type checker can ensure that units are consistent throughout the code.

For example, suppose you have a program that calculates the force exerted on an object using the equation F = m * a, where F represents force, m represents mass, and a represents acceleration. In a language that supports type checking with units, you can assign types to the variables, such as force: Newton, mass: Kilogram, and acceleration: MeterPerSecondSquared. The type checker can then verify that the units on both sides of the equation match, providing a form of formal verification.

By leveraging homogeneity through static type checking, software developers can catch potential errors, such as mixing incompatible units or performing incorrect calculations involving physical quantities. This approach enhances the reliability and correctness of software that deals with physical quantities.

It's important to note that formal software verification involves more than just ensuring homogeneity of physical quantities. It encompasses a broader range of techniques and methods to rigorously analyze and prove properties about software systems. However, leveraging the concept of homogeneity from the SI system can be a valuable tool in the pursuit of formal software verification, specifically when dealing with physical quantities.

The approach of leveraging the homogeneity of physical quantities and applying formal software verification techniques can significantly contribute to ensuring functional and high-quality software for IoT (Internet of Things) and Cyber-Physical Systems (CPS). Here's how it can be beneficial:

Correctness and Safety: IoT and CPS often involve complex interactions between physical components and software systems. Verifying the correctness and safety of the software controlling these systems is crucial to prevent potential hazards or malfunctions. By using formal software verification techniques, including enforcing homogeneity, developers can catch errors early in the development process and ensure that the software behaves as intended, reducing the risk of system failures or safety incidents.

Consistency of Units: In IoT and CPS, physical quantities are frequently exchanged between different components or subsystems. Ensuring the consistency of units throughout the system is vital to prevent integration issues and misinterpretation of data. By enforcing homogeneity in the software code, errors related to unit conversions or mismatched units can be detected statically, reducing the chances of introducing unit-related bugs during system integration.

Interoperability: IoT systems often involve diverse devices and platforms that need to interact seamlessly. By using a consistent approach to handling physical quantities, based on the SI system and homogeneity, software components can exchange and interpret data consistently, regardless of the specific devices or platforms involved. This promotes interoperability and facilitates the integration of heterogeneous systems within IoT and CPS environments.

Maintenance and Evolvability: IoT and CPS systems are typically subject to updates, maintenance, and evolution over their lifecycle. Enforcing homogeneity and applying formal verification methods can enhance software maintainability and evolvability. By establishing clear and consistent units and enforcing them in the code, developers can more easily understand and modify the software, reducing the risk of introducing errors during updates or modifications.

Quality Assurance: Formal software verification methods, including the use of homogeneity, contribute to a rigorous quality assurance process. By systematically applying verification techniques, developers can identify and eliminate potential software defects, improving the overall quality and reliability of IoT and CPS systems. This, in turn, enhances user satisfaction, reduces the risk of failures, and increases confidence in the deployed software.

By combining the principles of homogeneity from the SI system with formal software verification methods, developers can create more robust, reliable, and functional software for IoT and CPS. This approach helps mitigate risks, ensure safety, enhance interoperability, facilitate maintenance, and improve the overall quality of the software deployed in these systems.

Point 2: You mentioned that the proposed method cannot check expressions with angles, angle speed, and similar features. Could you elaborate on the transformation for physical value orientation introduced by Siano and how it enables the verification of software code in these cases? What are the key considerations and challenges in incorporating orientational homogeneity into the software verification process?

Response 2: We would like to emphasize that existing methods based solely on dimension analysis are unable to effectively check expressions involving angles, angular speed, and similar features. However, we have proposed a novel method that combines both dimension analysis and orientation analysis. This method enables us to effectively check expressions involving angles, angular speed, angular accelerations, and other similar features.

Incorporating orientational homogeneity, which refers to consistency in the orientations or directions of physical quantities, into the software verification process presents some key considerations and challenges. Here are a few important aspects to consider:

Representation of Orientations: Physical quantities that have orientations, such as vectors or tensors, require appropriate representation in software code. Choosing a suitable data structure or abstraction to represent orientations is crucial. This involves deciding on the coordinate system, defining the necessary operations and transformations, and ensuring consistency in the representation across different parts of the codebase.

Type Systems and Static Analysis: Incorporating orientational homogeneity often requires extending the type system or static analysis capabilities of programming languages. The type system needs to handle orientation-specific operations and enforce consistency in operations involving orientations. This may involve defining new type rules, constraints, or extensions to the type system to capture the orientations of quantities accurately.

Orientation Inference: In some cases, the orientation of a physical quantity may not be explicitly provided but needs to be inferred based on the context or other known information. Developing algorithms or techniques for inferring orientations from available data or constraints is a challenge that needs to be addressed to ensure the correctness and consistency of orientational homogeneity.

Dimensional Analysis: In addition to orientational homogeneity, dimensional analysis plays a crucial role in verifying software involving physical quantities. Combining orientational and dimensional analysis to ensure both the correct units and orientations are consistent throughout the code adds complexity to the verification process. Coordinating these two aspects effectively requires careful consideration and integration of both dimensions and orientations.

Tooling and Libraries: Developing tools, libraries, or frameworks that provide support for orientational homogeneity can greatly assist software verification efforts. Creating or adapting existing static analysis tools, type checkers, or formal verification frameworks to handle orientations and their consistency helps developers in enforcing orientational homogeneity more effectively and efficiently.

Complexity and Scalability: Incorporating orientational homogeneity into the software verification process introduces additional complexity to the analysis and verification techniques. As the complexity of the system increases, the scalability of the verification process becomes a challenge. Ensuring that the verification techniques can handle larger systems, complex interactions, and real-world scenarios while maintaining reasonable performance is an ongoing challenge.

Overall, incorporating orientational homogeneity into the software verification process requires careful consideration of representation, type systems, inference techniques, dimensional analysis, tooling, and managing complexity. Addressing these considerations and challenges is essential for effectively enforcing orientational homogeneity and ensuring the correctness and quality of software involving physical quantities with orientations.

Point 3: How did you develop the special software defect models based on the statistical characteristics of software code? Could you provide some insights into the statistical analysis tool used and the specific characteristics analyzed in the 2 GB of C++ GITHUB code for drones? What were the main findings or observations from this analysis?

Response 3: To develop the special software defect models based on the statistical characteristics of software code, we followed a systematic approach. Here are the general steps involved:

  1. Data Collection: We collected a substantial amount of software code for analysis. In our case, the source code of Unmanned Aerial Vehicle Systems from GitHub, totaling 2 GB in volume and consisting of 20,000 files was analyzed.
  2. Statistical Analysis: We used a special elaborated Statistical Analyzer of C++ to analyze the collected software code. This tool has 164 KB of C# source code.

This tool involved examining various statistical characteristics, such as the total number of variables per file, their distribution, probabilities of operations and variables, and other relevant metrics. This tool generate visual representations of the statistical characteristics using techniques like histograms and probability density functions. These visualizations helped in understanding the distribution patterns and trends in the data.

3 Identification of Defect Patterns: By analyzing the statistical characteristics, we looked for patterns and anomalies that could indicate potential software defects based on uncorrected both dimension and orientation homogeneity. For Identification of Defects we used the probability tries. The probabilities of both software operations and variables was used for development of models.

4 Model Development: Based on the identified defect patterns, we developed specialized software defect models. These models based both dimension and orientation homogeneity., enabling them to effectively check expressions involving dimensions, angles, speed, acceleration, and other relevant features.

5 Model Evaluation and Refinement: The developed models were evaluated using various techniques, such as manual  testing them on known defective code samples. Feedback and insights from these evaluations were used to refine and improve the models further.

It's important to note that statistical defect prediction models are not perfect and have limitations. The accuracy and effectiveness of the models depend on the quality and representativeness of the data, the choice of features, and the specific context in which they are applied. Nonetheless, statistical defect prediction can be a valuable approach to augmenting software development efforts and improving software quality.

Point 4: In your evaluation of the proposed method, you mentioned that it can detect over 60% of latent software defects based on the actual distribution of software characteristics. Can you discuss the implications of this detection rate and how it compares to existing software verification methods? What are the potential benefits for reducing testing time, improving reliability, and enhancing software quality?

Response 4: A detection rate of over 60% for latent software defects based on the proposed method and the actual distribution of software characteristics can have significant implications for software development and verification. Here are some points to consider:

1 Early Detection: The detection rate of over 60% implies that the proposed method is effective in identifying a substantial portion of latent defects before they manifest as issues or bugs during runtime or in production. Early detection allows developers to address these defects during the development process, reducing the likelihood of defects reaching end-users and potentially causing system failures or malfunctions.

2 Reduced Testing Time: Traditional testing methods often require exhaustive test case generation and execution, which can be time-consuming. The proposed method, on the other hand, uses statistical analysis to identify defect patterns, allowing for targeted testing on specific areas of the code. This targeted approach can significantly reduce testing time by prioritizing high-risk areas.

3 Cost Savings: Detecting a significant percentage of latent defects can lead to cost savings in terms of both time and resources. Addressing defects during the development phase is generally less time-consuming and expensive compared to fixing them in production. By catching a large number of defects early on, the method can help minimize the need for extensive debugging, maintenance, or post-release updates, resulting in overall cost reduction.

4 Improved Software Quality: The ability to detect over 60% of latent defects suggests that the proposed method contributes to enhancing software quality. By identifying and resolving these defects proactively, developers can improve the reliability, robustness, and performance of the software. This, in turn, enhances user satisfaction, minimizes disruptions, and builds trust in the software.

5 Complementary Approach: Existing software verification methods typically include a combination of techniques such as static analysis, dynamic testing, formal methods, and code reviews. The proposed method can be seen as a complementary approach that focuses on detecting latent defects based on software characteristics. By incorporating this method into the existing verification toolbox, developers can benefit from an additional layer of defect detection and improve overall verification effectiveness.

6 Comparative Analysis: To assess how the proposed method compares to existing software verification methods, it's crucial to consider their respective strengths, limitations, and applicability. The detection rate of over 60% provides a basis for comparison against the performance of other methods. Evaluating the proposed method alongside existing techniques in terms of detection rates, false positive rates, scalability, ease of integration, and resource requirements can provide insights into its comparative advantages and areas where it complements or surpasses existing approaches.

7 Continuous Improvement: The detection rate achieved by the proposed method serves as a baseline for further improvement and refinement. Ongoing research and development can focus on enhancing the method's accuracy, expanding its capabilities, addressing any limitations, and refining the models and algorithms used. Continuous improvement efforts can contribute to even higher detection rates, leading to further advancements in software quality assurance.

Overall, the high detection rate of the proposed method and its potential benefits in reducing testing time, improving both reliability and software quality demonstrate its value as an efficient and effective approach to software defect detection.

Point 5: How scalable is the proposed software verification method? Does it have any limitations or considerations when applied to larger software systems or more complex IoT and Cyber-Physical Systems? Are there any specific factors or characteristics that may impact the effectiveness of the method in different contexts?

Response 5: For compile-time software verification, the limitations of the proposed software verification method are equal to the limitations of the compiler used. However, for runtime software verification, the limitations of the proposed method do not solely depend on compiler limitations. As the codebase grows, the verification process may indeed require more time and computational resources. Nevertheless, to address scalability challenges, techniques such as modular analysis or sampling strategies can be employed to focus on specific modules or representative subsets of the codebase. These techniques help optimize the verification process and improve scalability.

There are specific factors or characteristics that may impact the effectiveness of the method:

  1. Unknown dimensions and orientations of input and output parameters of systems, units, statements.
  2. Probability distribution of operations.
  3. Probability distribution of variables according to their dimensions and orientations.

Point 6: Can you provide some examples or scenarios where the proposed software verification method has been applied or could be applied in practice? How does it fit into the overall software development lifecycle for IoT and Cyber-Physical Systems? Are there any specific industries or domains where this method could have significant impact or benefits?

Response 6: The proposed software verification method can be applied in various scenarios and domains within the software development lifecycle for IoT and Cyber-Physical Systems. Here are some examples:

  1. Automotive Industry: The method can be applied to verify software components in autonomous vehicles, ensuring their safety and reliability. By detecting latent defects based on statistical characteristics, the method can help prevent potential system failures that could lead to accidents.
  2. Healthcare: In the development of medical devices and healthcare software, the method can be used to verify critical functionalities and ensure patient safety. By identifying and addressing potential defects early on, the method can contribute to the delivery of high-quality and reliable healthcare systems.
  3. Industrial Automation: In industrial settings, the method can verify software used in control systems and manufacturing processes. By detecting defects that could cause equipment failures or production issues, the method can help optimize operations and minimize downtime.
  4. Smart Grids: The method can be applied to verify software components in smart grid systems, which involve the integration of power grids with advanced communication and control technologies. By ensuring the reliability and security of the software, the method can contribute to efficient energy management and grid stability.
  5. Internet of Things (IoT): The method can be utilized in the development of IoT applications and systems. By verifying the software that controls IoT devices and the interactions between them, the method helps ensure the functionality, security, and interoperability of the IoT ecosystem.

In terms of its fit into the overall software development lifecycle, the method can be integrated at different stages, including design, implementation, and testing phases. It can complement existing verification techniques, such as static analysis, testing, and code reviews, by providing an additional layer of defect detection. By incorporating the method into the development lifecycle, potential defects can be identified and addressed early on, improving software quality and reducing the likelihood of issues during deployment and operation.

The impact and benefits of the method can vary across industries and domains. Industries that rely heavily on software systems with critical functionalities, safety requirements, or regulatory compliance, such as automotive, aerospace, healthcare, and energy sectors, can benefit significantly from the method's ability to detect latent defects and enhance software reliability. However, the method's applicability is not limited to these industries, and it can be adapted to various other domains where software plays a crucial role in system operation and performance.

Overall, the proposed software verification method has the potential to positively impact software development in IoT and Cyber-Physical Systems by improving software quality, enhancing reliability, and reducing risks associated with latent defects.

Point 7: Based on your research, what are the key insights or recommendations for software developers or organizations looking to improve the verification of IoT and Cyber-Physical Systems software? Are there any specific best practices or guidelines that you would suggest based on the findings of your study?

Response 7: There are some key insights and recommendations for software developers or organizations looking to improve the verification of IoT and Cyber-Physical Systems software:

  1. Incorporate Multiple Verification Techniques: Adopt a combination of verification techniques, such as static analysis, testing, formal methods. Each technique has its strengths and limitations, and by leveraging multiple approaches. The combination can enhances the effectiveness of software verification and increases the chances of detecting defects.
  2. Consider Dimension and Orientation Homogeneity: Integrate dimension and orientation homogeneity as a key aspect of software verification for IoT and Cyber-Physical Systems. By ensuring consistency and compatibility in the physical quantities and units used in the software, we can minimize errors and improve the overall quality and reliability of the system.
  3. Focus on Early Detection: Emphasize early defect detection during the software development lifecycle. Detecting and addressing defects at early stages reduces the cost and effort of fixing them later. Integrate verification activities throughout the development process, including design reviews, code inspections, and testing, to identify and mitigate defects as early as possible.
  4. Establish Testing and Verification Standards: Define and enforce testing and verification standards specific to IoT and Cyber-Physical Systems software. This includes establishing guidelines for unit testing, integration testing, system testing, and validation. Standardized practices help ensure consistency and reliability across different software components and facilitate easier verification.
  5. Use Realistic Test Environments: Create test environments that closely resemble the operational environment of the IoT or Cyber-Physical Systems. This includes simulating real-world conditions, incorporating relevant sensors, emulating network interactions, and considering environmental factors. Realistic testing environments increase the chances of uncovering potential issues and improving the system's robustness.
  6. Continuous Monitoring and Improvement: Implement continuous monitoring of the software performance and conduct regular audits to identify potential defects or anomalies. Monitor system behavior. Continuously improve the verification process based on the insights gained from monitoring and analysis.

Point 8: Are there any potential future developments or extensions to the proposed method that you are currently exploring or would like to investigate? Are there any open research questions or areas where further research is needed to enhance or refine the software verification process for IoT and Cyber-Physical Systems?

Response 8: There are some potential future developments and open research questions to enhance and refine the software verification process for IoT and Cyber-Physical Systems:

  1. Integration of Physical Modeling: Investigate the integration of physical modeling techniques into the software verification process. This involves capturing the physical behavior of the system, such as its dynamics, constraints, and interactions with the environment, and using this information to enhance the verification accuracy and fidelity.
  2. Runtime Monitoring and Adaptation: Research how runtime monitoring and adaptation can be utilized to enhance software verification. Develop approaches that dynamically monitor system behavior, detect anomalies or deviations from expected models, and trigger adaptive actions to maintain system integrity and reliability.
  3. Verification of Machine Learning Components: Investigate techniques to verify software components that utilize machine learning algorithms, such as neural networks, in IoT and Cyber-Physical Systems. This includes methods to verify the correctness, robustness, and safety of these components and ensure their reliable integration within the overall system.
  4. Cross-Layer Verification: Explore verification techniques that span multiple layers of the system stack, including hardware, communication protocols, and software. Investigate approaches to ensure compatibility, integrity, and security across different layers, considering the tight integration and interdependencies in IoT and Cyber-Physical Systems.
  5. Formal Methods for Cybersecurity Verification: Develop formal methods and verification techniques specifically tailored for cybersecurity verification in IoT and Cyber-Physical Systems. This includes analyzing system vulnerabilities, identifying potential attack vectors, and verifying the effectiveness of security measures.
  6. Verification of System-of-Systems: Investigate verification methods for large-scale systems composed of multiple interconnected subsystems, known as system-of-systems. Develop techniques to ensure compatibility, interoperability, and overall system behavior in complex and distributed environments.
  7. Standards and Certification: Research the development of industry standards and certification processes specifically addressing software verification for IoT and Cyber-Physical Systems. Establish guidelines and best practices to ensure the safety, security, and reliability of these systems in different application domains.

Point 9: How does your software verification method integrate with other existing software engineering practices, such as testing, code review, or formal methods? Can you discuss the complementarity or synergies between your approach and these established techniques?

Response 9: The proposed software verification method can integrate with other existing software engineering practices, such as testing, code review, and formal methods. There are complementarities and synergies between the proposed approach and these established techniques:

  1. Testing: Software testing is a widely used practice for identifying defects and ensuring software quality. The proposed verification method can complement testing by providing an additional layer of defect detection. The combination of testing and statistical defect prediction can increase the effectiveness and coverage of defect detection.
  2. Code Review: Code reviews involve manual inspection of software code by developers or peers to identify defects, improve code quality, and ensure adherence to coding standards. The proposed verification method can support code review by providing quantitative metrics. These metrics can help reviewers focus their attention on critical areas that have a higher probability of containing defects.
  3. Formal Methods: Formal methods, such as model checking or theorem proving, use mathematical techniques to rigorously analyze and verify software behavior. The proposed verification method can be complementary to formal methods.

Point 10: What are the main contributions or novel aspects of your research in the context of software verification for IoT and Cyber-Physical Systems? How does your work advance the current state of the art and contribute to the broader field of software engineering and system reliability?

Response 10: The proposed method for software verification in the context of IoT and Cyber-Physical Systems makes several contributions and introduces novel aspects that advance the current state of the art and contribute to the broader field of software engineering and system reliability:

  1. Dimension and Orientation Homogeneity: The incorporation of dimension and orientation homogeneity as a fundamental aspect of software verification is a novel contribution. By ensuring consistency and compatibility in the physical quantities and units used in the software, the proposed method addresses a critical aspect of IoT and Cyber-Physical Systems where precise and accurate representation of physical quantities is essential. This contributes to the overall system reliability and prevents errors arising from incompatible units or dimensions.
  2. Statistical Characteristics and Defect Prediction: The proposed method leverages statistical characteristics of software code to predict the presence of defects. This approach introduces a quantitative and data-driven aspect to the software verification process. By analyzing the statistical distribution of software characteristics, such as dimensions and orientations of parameters, variables, and operations, the method identifies patterns and correlations that can indicate the presence of defects. This statistical defect prediction is a novel contribution that enhances the ability to detect and address defects in IoT and Cyber-Physical Systems.
  3. Practical Applicability and Integration: The proposed method is designed with practical applicability in mind. It can be integrated into existing software engineering practices, such as testing, code review, and formal methods, without requiring significant changes or disruptions to the development process. This practicality ensures that the method can be readily adopted and applied in real-world scenarios, contributing to the broader field of software engineering and system reliability by providing a practical and effective approach to software verification for IoT and Cyber-Physical Systems.
  4. Industry Relevance: The proposed method addresses the specific challenges and requirements of IoT and Cyber-Physical Systems, which are increasingly important in various industries, including automotive, healthcare, manufacturing, and smart infrastructure. By focusing on the verification of software in these domains, the method offers practical benefits and advancements that are directly relevant to industries where system reliability, safety, and performance are critical.

Overall, the contributions and novel aspects of the proposed method lie in its incorporation of dimension and orientation homogeneity, statistical defect prediction, practical applicability, and relevance to the industry. These advancements advance the current state of the art in software verification for IoT and Cyber-Physical Systems and contribute to the broader field of software engineering and system reliability by providing an effective, data-driven, and practical approach to ensure the quality and reliability of software in these complex and interconnected systems.

Also, additional text has been inserted in the introduction (see lines 762-765 in the new version of the manuscript) and conclusions (see lines 160-273 in the new version of the manuscript).

Please see the attachment new version of manuscript

Thank you for your time!

Reviewer 2 Report

Paper presents a novel technique for software verification, which appears to have considerable merit. My comments are included as a markup to the pdf of the paper.

English is understandable but cleanup is needed.

Author Response

Response to Reviewer 2 Comments

Point 1: Sentence fragment: “CPS similar to the IoT”

Response 1: We have made the requested change by updating the text (see lines 37-43 in the new version of the manuscript).

Point 2: I cannot discern this observation from Figure 1: “However, IoT Analytics has revised its growth outlook for 2023 to 18.5% 66 (down from 24% previously) [7], as shown in Figure 1”.

Response 2: We have taken note of your response, and we have made the requested change by updating the text (see lines 69-72 in the new version of the manuscript).

Point 3: Popularity rate is a vague metric. Are you saying that 56.9% of IoT developments in 2022 used C/C++?

Response 3: We have acknowledged your response, and we have made the requested change by updating the text (see lines 79-97 in the new version of the manuscript).

Point 4: Actually testing is one method of verification. And it is not the only method. And testing also is conducted outside the context of verification. Any good sw engineering text can provide you with the correct nomenclature here. “This makes program verification an alternative or complement to testing”

Response 4: We have noted your response, and we have made the requested change by deleting the mentioned sentence. The text has been updated accordingly (see lines 143-152 in the new version of the manuscript).

Point 5: Perhaps you will explain later in the paper, but it is not evident to me how this approach would be useful for software verification if physical calculations and measurements are not included in a particular application. Granted, CPS systems by their nature tend to include such quantities, but that is less prevelant for the IoT in general.

Response 5: We have taken note of your response, and we have made the requested change by inserting the additional text in the manuscript (see lines 160-273 in the new version).

Point 6: Reference needed “Siano proposed an orientational analysis (OA) that involves extending physical dimensions to improve DA.”

Response 6: We have taken note of your response, and we have made the requested change by inserting the reference in the text (see lines 363-365 in the new version of the manuscript).

Point 7: Exactly. Use of DA is not sufficient for SW verification. “Therefore, while dimensional analysis can help identify some potential issues with the equation, it may not be able to catch all possible defects. ”

Response 7: We agree with your response, and we have made the necessary changes to the text by inserting the additional text (see lines 486-487 in the new version of the manuscript).

Point 8: Will you get into the implications of this assumption? “Despite the presence of these two types of defects, the model still assumes that only one defect is present in the system at any given time. ”

Response 8: We agree with your response, and we have made the appropriate changes to the text (see lines 523-524 in the new version of the manuscript).

Point 9: If I followed this correctly, you are confusing the actual detection of defects with the probability of detecting defects.

Response 9: We agree with your warning, and we have made the necessary changes to the text (see lines 739-742 in the new version of the manuscript).

Point 10: Per my previous comment, as written it appears that you are developing a model which has a probability of detecting defects. And the last sentence of this paragraph is written accordingly. However, the fist sentence in the paragraph implies actual defect detection.

Response 10: We agree with your warning. The text has been changed accordingly (see lines 762-765 in the new version of the manuscript).

Also, additional text has been inserted in the introduction (see lines 762-765 in the new version of the manuscript) and conclusions (see lines 160-273 in the new version of the manuscript). 

Please find attached the new version of the manuscript.

Thank you for your time!

Round 2

Reviewer 1 Report

The author address all the previous comments but still some issues are pending:

The abstract needs to be rewritten to point out significance and impact of the paper.

In the related work, it is recommended to refer the contribution made by the researchers and the novelty of the research. However, the author does not mention that.

I recommend that the authors add some more current articles to improve the paper's overall quality. The preparation of a comparative analysis of the current publications on this subject should also be included.

Avoid presenting with lengthy paragraph.

Paper needs to polish and provide a detailed explication of theoretical/systematic aspects behind this paper.

Some more clarification regarding the motivation and challenges of the proposed approach, and how the prescribed scheme would overcome them must be added in the revised version

Notations and acronyms used in this paper should be summarized in a table to organize this paper in a better way.

Improve the quality of figures and explain those properly.

Finally, a final proof-reading is highly suggested, in order to correct some typos.

Minor editing of English language required

Author Response

Response to Reviewer 1 Comments

Point 1: • The abstract needs to be rewritten to point out significance and impact of the paper.

Response 1: According to your warning, the abstract of our paper was rewritten.

Point 2: • In the related work, it is recommended to refer the contribution made by the researchers and the novelty of the research. However, the author does not mention that.

Response 2: According to your warning, the contributions made by the researchers were inserted in the text.

Point 3: • I recommend that the authors add some more current articles to improve the paper's overall quality. The preparation of a comparative analysis of the current publications on this subject should also be included.

Response 3: According to your warning, current articles were used to improve our paper. The comparative analysis of recent publications on modern software tools was included in the paper.

Point 4: • Avoid presenting with lengthy paragraph.

Response 4: According to your warning, the lengthy paragraphs were rewritten.

Point 5: • Paper needs to polish and provide a detailed explication of theoretical/systematic aspects behind this paper.

Response 5: According to your warning, a detailed explanation of the theoretical/systematic aspects behind this paper was included, and our paper was rewritten.

Point 6: • Some more clarification regarding the motivation and challenges of the proposed approach, and how the prescribed scheme would overcome them must be added in the revised version

Response 6: According to your warning, more clarification regarding the motivation and challenges of the proposed approach was included.

Point 7: • Notations and acronyms used in this paper should be summarized in a table to organize this paper in a better way.

Response 7: According to your warning, most of the special acronyms were deleted from the text.

Point 8: • Improve the quality of figures and explain those properly.

Response 8: According to your warning, the quality of the figures was improved, and explanations were added to the figures.

Point 9: • Finally, a final proof-reading is highly suggested, in order to correct some typos.

Response 9: According to your warning, all typos detected in the paper were corrected.

Thank you for your time!